# Progressive Token Length Scaling in Transformer Encoders for Efficient Universal Segmentation

**Abhishek Aich**[1][*]**, Yumin Suh**[4][†] **, Samuel Schulter**[3][†]**, Manmohan Chandraker**[1,2]
[1]NEC Laboratories, America, USA,
[2]University of California, San Diego, USA, [3]Amazon AGI, [4]Atmanity Inc.

## Abstract

A powerful architecture for universal segmentation relies on transformers that encode multi-scale image features and decode object queries into mask predictions. With efficiency being a high priority for scaling such models, we observed that the state-of-the-art method Mask2Former uses $>$50% of its compute *only* on the transformer encoder. This is due to the retention of a full-length token-level representation of all backbone feature scales at each encoder layer. With this observation, we propose a strategy termed **PRO**gressive Token Length **SCAL**ing for **E**fficient transformer encoders (`PRO-SCALE`) that can be plugged-in to the Mask2Former segmentation architecture to significantly reduce the computational cost. The underlying principle of `PRO-SCALE` is: progressively scale the length of the tokens with the layers of the encoder. This allows `PRO-SCALE` to reduce computations by a large margin with minimal sacrifice in performance ($\sim$52% encoder and $\sim$ 27% overall GFLOPs reduction with *no* drop in performance on COCO dataset). Experiments conducted on public benchmarks demonstrates `PRO-SCALE`'s flexibility in architectural configurations, and exhibits potential for extension beyond the settings of segmentation tasks to encompass object detection. Code available here: https://github.com/abhishekaich27/proscale-pytorch

## 1 Introduction

The tasks of image segmentation (instance (He et al., 2017), semantic (Tu, 2008), and panoptic (Kirillov et al., 2019)) are recently being addressed together under the paradigm of "universal" image segmentation (Cheng et al., 2021; 2022; Jain et al., 2023; Gu et al., 2024; Cavagnero et al., 2024; Rosi et al., 2024). This is due to the evolution of transformer-based (Vaswani et al., 2017) approaches that can represent both *stuff* and *things* categories (Kirillov et al., 2019)) using general tokens, leading to a diminished distinction among the tasks of semantic, instance, and panoptic segmentation. Success of the state-of-the-art universal segmentation framework Mask2Former (M2F) (Cheng et al., 2022) can be attributed to its DEtection TRansformer (Carion et al., 2020) or DETR-style architecture. This DETR-style segmentation (henceforth termed as M2F-style) framework exhibits exceptional performance across various segmentation tasks *without* the need for task-specific design choices, setting them apart from preceding modern panoptic segmentation frameworks (Rashwan et al., 2024; Hu et al., 2023; Ammar et al., 2023; Sun et al., 2023).

The strong performance of M2F-style architecture, however, incurs significant computational overhead hindering their widespread deployment. In this paper, we address this important problem of designing an efficient M2F-style architecture for universal segmentation model. In particular, we present **PRO**gressive Token Length **SCAL**ing for **E**fficient transformer encoders or `PRO-SCALE` that reduces the computational load occurring in the transformer encoder of such models with surprisingly low performance deterioration. M2F-style architectures contain a *backbone* (for multiscale feature extraction from input images), a *pixel decoder* or transformer encoder (to capture long-range dependencies and contextual relationships across the multi-scale backbone features), and a *transformer decoder* (for predicting the masks and labels) along with a segmentation module (see

---

[*]Corresponding author, email: aaich001 *at* ucr *dot* edu; [†]Work done by YS and SS while at NECLA.

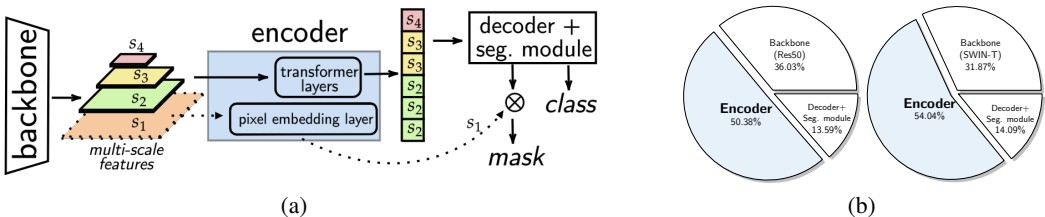

Figure 1: **Compute distribution.** *(a)* Mask2Former-style segmentation model *(b)* In the Mask2Former model using Res50 (He et al., 2016) and SWIN-T (Liu et al., 2021) backbones, the transformer encoder contributes the most to the overall computation cost, accounting for 54.04% and 50.38%, respectively.

Fig. 1(a)). In efficient segmentation models, where the backbones are traditionally lightweight (SWIN-T (Liu et al., 2021) and Res50 (He et al., 2016)), cross-scale feature attention has been shown as a potent way to achieve high segmentation performance (Cheng et al., 2022; Li et al., 2023a). However, it has been shown in (Li et al., 2023a; Lv et al., 2023; Yao et al., 2021) that due to a large number of query tokens introduced from multi-scale features of light backbones, the encoder incurs the *highest* computational cost. We make a similar observation in Fig. 1(b) for M2F where more than 50% of the compute comes from the encoder, given backbones SWIN-T and Res50.

Existing solutions (Li et al., 2023a; Lv et al., 2023) for making DETR-style architectures efficient are *only* designed to handle *detection* tasks. For example, Lite-DETR (Li et al., 2023a) proposed to update larger and smaller scale features in different frequencies for efficient computation. RT-DETR (Lv et al., 2023) on the other hand, proposed a hybrid encoder that transforms multi-scale features into a sequence of image features through intra-scale interaction and cross-scale fusion. Such detection-based strategies do not suit the segmentation task: either they propose to discard the use of multi-scale features (as in (Lv et al., 2023)) or lack the ability to reduce computations induced due to constructing a pixel embedding map (as in (Li et al., 2023a)). This significantly reduces their impact on performance-efficiency trade-off for segmentation models (*e.g.* ~33% GFLOPs reduction but 11% segmentation accuracy drop).

To this end, we introduce `PRO-SCALE` for M2F's transformer encoder that has two key properties: (*1*) progressively increase the token length or input size at each encoder layer by introducing larger scale features in the deeper layers (*2*) simplifying the pixel embedding layer by replacing it with a *Light Pixel Embedding* (LPE) module. The fundamental idea of `PRO-SCALE` is to address the possible redundancy that arises from consistently maintaining a *full-length token sequence* across all layers in the encoder. For example as shown in Fig. 2(a), M2F uses tokens from multi-scale features across *all* encoder layers resulting in expensive computations. With `PRO-SCALE`'s progressively expanding tokens, the reduction in sequence length leads to significant FLOPs savings with minimal to negligible degradation in segmentation performance (see Fig. 2(b)). Extensive experiments show that `PRO-SCALE` based M2F architecture achieves a ~52% reduction in transformer encoder GFLOPs while maintaining the same segmentation performance. In particular, `PRO-SCALE`-M2F with SWIN-T backbone achieves a 52.82% PQ with 171.7 GFLOPs (*vs.* original performance of 52.03% PQ with 234.5 GFLOPs) on the COCO (Lin et al., 2014) dataset.

To summarize, we present an efficient transformer encoder `PRO-SCALE` for M2F universal segmentation architecture. `PRO-SCALE` operates on the fundamental idea of progressively expanding tokens along the encoder depth to address the computation redundancy. It is further assisted by a light pixel embedding module that effectively target computational reduction in the encoder module. Extensive experiments show that `PRO-SCALE` achieves the best performance-efficiency trade-off on two datasets across diverse settings.

## 2 RELATED WORKS

DETR (Carion et al., 2020) embraces an end-to-end object detection approach with a set-prediction objective, discarding the need for manually crafted modules like anchor design and non-maximum

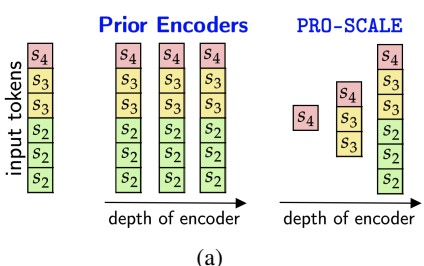
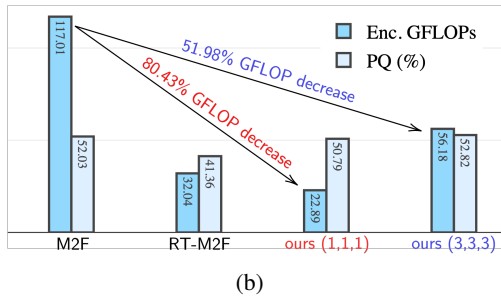

(a)  (b)

Figure 2: **Key idea and performance comparison of** PRO-SCALE **w.r.t. prior works.** *(a)* illustrates the key idea of PRO-SCALE to progressively extend the token length in the transformer encoder. $\{\mathbf{s}_2, \mathbf{s}_3, \mathbf{s}_4\}$ represent different resolutions. In *(b)*, we show two instantiates of our proposed transformer encoder PRO-SCALE, compared with Mask2Former (M2F) (Cheng et al., 2022) and RT-M2F (an adaptation of (Lv et al., 2023)). PRO-SCALE eliminates 80.43% (with configuration $(p_1, p_2, p_3) = (1,1,1)$) and 51.98% (with configuration $(p_1, p_2, p_3) = (3,3,3)$) of encoder GFLOPs from M2F while maintaining the competitive performance. Results are computed on the COCO (Lin et al., 2014) dataset.

suppression. Using the set prediction mechanism introduced in DETR, MaskFormer (Cheng et al., 2021) proposed an approach that converts any existing per-pixel classification model into a mask classification model. This resulted in a universal segmentation architecture that demonstrated state-of-the-art performance across all segmentation tasks on public benchmarks in diverse settings (Cheng et al., 2022; Li et al., 2023b; Ding et al., 2022; Cavagnero et al., 2024; Rosi et al., 2024) without task-specific design choices. In particular, it employs a transformer decoder to predict a set of pairs (a class prediction, a binary mask). Mask2Former (Cheng et al., 2022) improved Mask-Former by using *masked* attention in the transformer decoder to restrict the attention to localized features centered around predicted segments. Recently, (Li et al., 2023b) presented a DETR-style multi-task architecture by extending DINO (Zhang et al., 2022) for both detection and segmentation tasks. PEM (Cavagnero et al., 2024) introduced a MaskFormer based model that includes a multi-scale feature pyramid network to extract high-semantic-content features with context-based self-modulation to ensure efficiency. PRO-SCALE focuses on making the Mask2Former universal segmentation model efficient.

In recent times, there have been some interesting efforts (Wang et al., 2019; de Geus et al., 2020; Hong et al., 2021; Hou et al., 2020; Sun et al., 2023; Šarić et al., 2023; Cavagnero et al., 2024; Xu et al., 2024; Rosi et al., 2024) to make task-specific or *non-M2F-style* panoptic segmentation models efficient. For example, YOSO (Hu et al., 2023) introduced a lightweight panoptic segmentation model by utilizing a feature pyramid aggregator and separable dynamic decoders. ReMaX (Sun et al., 2023) proposed a training pipeline to address the dominant impact of false positive mask assignments in panoptic segmentation by utilizing a separate semantic prediction head. Above methods contain specific model design choices or training strategies suited for the panoptic segmentation task. Unlike these, PRO-SCALE does not conform to one segmentation task and still shows competitive performances.

## 3 METHODOLOGY

**Framework Overview.** Following M2F, our proposed framework is composed of a lightweight backbone, our novel transformer encoder design PRO-SCALE, and a transformer decoder with mask and class prediction heads (each described next). The overall model framework is shown in Fig. 3. In particular, the input image is fed to the backbone to create multi-scale features. These features are flattened and updated by PRO-SCALE (introduced in Sec. 3.1) with gradual expansion of input tokens along with depth of intermediate encoder layers, allowing a better performance-efficiency trade-off. The transformer decoder uses the representations from PRO-SCALE and learnable object queries to compute mask embeddings. Finally, a segmentation module uses the decoder output and per-pixel embeddings from PRO-SCALE for mask predictions. PRO-SCALE provides these per-pixel embeddings required for mask prediction using a parameter-free *Light-Pixel Embedding*

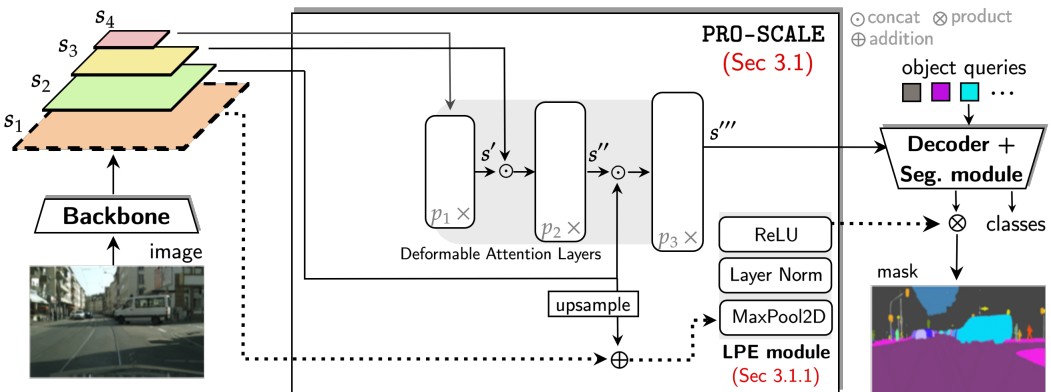

Figure 3: **Proposed framework.** Our model includes our transformer encoder PRO-SCALE (Sec. 3.1), designed to reduce the computational load. $\{s_i\}$s represent the multi-scale backbone features. PRO-SCALE progressively scale the length of the tokens with the layers of the encoder. This allows PRO-SCALE to reduce computations by a large margin with minimal sacrifice in performance. Further, $s_1$ goes through our efficient *Light-Pixel Embedding* (LPE) module (Sec. 3.1.1) to create pixel embeddings for mask prediction. $p_1$, $p_2$ and $p_3$ represent encoder layer frequency. $\{s', s'', s'''\}$ represent the outputs of respective layers.

module (described in Sec. 3.1.1). We start by introducing notations and model components, before diving into the details.

- *A backbone network* that extracts features from an input image $\mathbf{I} \in \mathbb{R}^{H \times W \times 3}$. The backbone network can provide multi-scale feature maps $\{\bar{\mathbf{s}}_1, \bar{\mathbf{s}}_2, \bar{\mathbf{s}}_3, \bar{\mathbf{s}}_4\}$. The spatial resolutions are typically $1/4^2$, $1/8^2$, $1/16^2$, and $1/32^2$ of the input image, respectively. We will denote the token form of these features as $\mathbf{s}_1, \mathbf{s}_2, \mathbf{s}_3$, and $\mathbf{s}_4$, respectively.

- *A pixel decoder or transformer encoder*, that enhances the image features $\{\mathbf{s}_2, \mathbf{s}_3, \mathbf{s}_4\}$ and also creates a per-pixel embedding map from $\mathbf{s}_1$ and $\mathbf{s}_2$. The feature enhancement is performed on a token representation $\mathbf{P} \in \mathbb{R}^{K \times C}$, where $K = \sum(HW/32^2 + HW/16^2 + HW/8^2)$ and $C = 256$ (Cheng et al., 2022). This $\mathbf{P}$ is composed of concatenated $\{\mathbf{s}_2, \mathbf{s}_3, \mathbf{s}_4\}$ obtained *via* flattening the spatial dimensions and fed to a transformer encoder. The transformer encoder usually consists of several stacked transformer blocks. Following M2F, our framework also consists of six deformable attention transformer (Zhu et al., 2020) layers that includes a self-attention block and a feed-forward block. The per-pixel embedding map, on the other hand, is generated using convolutional layers $\mathbf{s}_1$ and $\mathbf{s}_2$, which enhance local spatial details to be used to create the segmentation mask.

- *A Transformer decoder along with segmentation module* that decodes binary masks from the modulated image features from the pixel decoder using a set of randomly initialized object queries. The transformer decoder consists of three stages, with each stage consisting of three transformer layers (Vaswani et al., 2017). Each layer includes a self-attention block, a cross-attention block, and a feed-forward block. Each layer only handles tokens of one scale of features for efficiency (Cheng et al., 2022).

As described above, the original transformer encoder results in significantly expensive computations from global attention mechanism, as it maintains the complete token length of $\mathbf{P}$ for all layers. Simply dropping larger scale features results in poor localization, especially in small objects (Li et al., 2023a). For example, removing features $\mathbf{s}_1$ and $\mathbf{s}_2$ results in a degradation of $\sim 5\%$ **AP** compared to original model. Moreover, the use of large-scale features to produce pixel embedding map in the convolutional layers require processing an enormous number of tokens, leading to an extremely high computational demand. We will now explain how our approach tackles these obstacles.

### 3.1 PRO-SCALE: PROPOSED TRANSFORMER ENCODER

**Intuition.** The bottleneck towards an efficient encoder are excessive large-scale features, where most of which are not informative but contain local details for different objects (Li et al., 2023a). In order to better handle their usage in the encoder, we can leverage the composition of multi-scale

features from the backbone: (*1*) The limited quantity of small-scale features (*e.g.* $\mathbf{s}_3, \mathbf{s}_4$) captures abundant semantics. (*2*) The larger quantity of large-scale features (*e.g.* $\mathbf{s}_1, \mathbf{s}_2$) captures crucial local features important for segmenting varying scale of objects. For example, assuming a SWIN-T backbone based M2F, $\{\mathbf{s}_2, \mathbf{s}_3, \mathbf{s}_4\}$ result in $\{76.19\%, 19.04\%, 4.76\%\}$ token contributions, respectively (also see our analysis in Fig. 9 in Appendix). Therefore, we propose to prioritize enrichment of tokens from small scale features earlier than the large scale features.

**Encoder structure.** We build upon deformable attention due to its linear complexity with the number of feature queries (represented by $\mathbf{P}$). We create 3 splits of $\mathbf{P}$ (following the three scales):

$$\mathbf{P}_1 = \mathcal{C}(\mathbf{s}_4) \in \mathbb{R}^{K_1 \times C}, \tag{1}$$

$$\mathbf{P}_2 = \mathcal{C}(\mathbf{s}', \mathbf{s}_3) \in \mathbb{R}^{K_2 \times C} \tag{2}$$

$$\mathbf{P}_3 = \mathcal{C}(\mathbf{s}'', \mathbf{s}_2) \in \mathbb{R}^{K_3 \times C} \tag{3}$$

Here, $\mathcal{C}(\cdot)$ denotes the concatenation operation along the token size dimension, $K_3 = \sum(^{HW}/_{32^2} + {^{HW}}/_{16^2} + {^{HW}}/_{8^2})$, $K_2 = \sum(^{HW}/_{32^2} + {^{HW}}/_{16^2})$, and $K_1 = {^{HW}}/_{32^2}$. These splits are then fed to three stages of the deformable attention transformer layers sequentially. The output of these stages are $\mathbf{s}', \mathbf{s}''$, and $\mathbf{s}'''$, respectively. $\mathbf{s}'''$ is subsequently fed to the decoder. Each stage is repeated $p_1, p_2$ and $p_3$ times before propagating the tokens to the next stage. This results in updation of $\mathbf{s}_4$ for $(p_1 + p_2 + p_3)$, $\mathbf{s}_3$ for $(p_2 + p_3)$, and $\mathbf{s}_2$ for $p_1$ times, gradually *expanding* the token length of inputs to intermediate layers in PRO-SCALE. As we show in Sec. 4, this strategy reduces computing load by significant margins while maintaining performance. Note that, since we skip using tokens from large scale features in the initial layers, we also use a token recalibration within PRO-SCALE that enriches small-scale features with high-scale features to gain slightly better segmentation without significant computational overhead. Please see Sec. C in Appendix for more details and Fig. 7 for a complete visualization.

### 3.1.1 LIGHT-PIXEL EMBEDDING (LPE) MODULE

**Intuition.** Strong performance of M2F depends on multi-scale features computed from the backbone (Cheng et al., 2022). Tokens $\{\mathbf{s}_2, \mathbf{s}_3, \mathbf{s}_4\}$ are fed to the encoder layers to compute $\mathbf{s}'''$ in order to produce per-segment embeddings in the transformer decoder. $\mathbf{s}_1$, on the other hand, serves the purpose of creating the per-pixel embedding map $\mathcal{E}_{emb}$ to enhance local details in the feature maps. However, due to the large size of $\mathbf{s}_1$, it results in high computational load from the use of convolutional layers in original M2F. Hence rather than dropping $\mathbf{s}_1$, we weaken this inductive bias and assess the existing learnable module with a simpler module in our design.

**LPE structure.** We propose to use a simple maxpooling layer followed by normalization and non-linearity to compute $\mathcal{E}_{emb}$. The goal of our LPE module is to mitigate this overhead while keeping the advantages of $\mathbf{s}_1$ in producing $\mathcal{E}_{emb}$. We observe it reduces almost $\sim 45\%$ GFLOPs in the encoder architecture, but doesn't significantly harm the overall model's segmentation performance. In our implementation, we use a pooling kernel size of 3 (and stride is set to 1).

## 4 EXPERIMENTS

In this section, we evaluate PRO-SCALE based M2F architecture on two benchmarks on all segmentation tasks. *First*, we observe that PRO-SCALE is extremely effective in reducing computational load while providing best trade-offs compared to baselines (Tab. 1, 2, 9). *Second*, we provide an extensive ablation analysis of PRO-SCALE in Tab. 3 - 7, and Fig. 4, 5. *Third*, we test the performance of PRO-SCALE on prevalent state-of-the-art frameworks for tasks like object detection, joint-task predictions, and open-vocabulary universal segmentation in Tab. 8, 10 and 11. *Finally*, we visualize some segmentation examples in Fig. 6. In the Appendix, we provide our training, and dataset (for COCO (Lin et al., 2014) and Cityscapes (Cordts et al., 2016)) details. For brevity, we will denote our overall framework as PRO-SCALE.

**Evaluation metrics.** We evaluate PRO-SCALE and baselines in the *universal* segmentation setting. This means, following (Cheng et al., 2022), PRO-SCALE's evaluation is performed using a model trained *exclusively* with panoptic segmentation annotations. For panoptic segmentation, the conventional **PQ** (Panoptic Quality (Kirillov et al., 2019)) metric is used. Following (Cheng et al.,

Table 1: **COCO evaluation.** PRO-SCALE (with configuration $(p_1, p_2, p_3)$) is extremely competitive against the baselines on COCO with at least 51.99% GFLOPs reduction compared to M2F with no performance drop. Non-M2F-style architectures are colored in gray.

| Model | Performance (↑) | | | GFLOPs (↓), (% decrement) | |
|---|---|---|---|---|---|
| | **PQ** | **mIOU**$_p$ | **AP**$_p$ | **Total** | **Encoder** |
| **Backbone: SWIN-T** | | | | | |
| M2F (Cheng et al., 2022) | 52.03 | 62.49 | 42.17 | 234.50 | 117.00 |
| Lite-M2F (Li et al., 2023a) | 52.70 | 63.08 | 41.10 | 188.00 (-19.83) | 79.78 (-31.81) |
| RT-M2F (Lv et al., 2023) | 41.36 | 61.54 | 24.68 | 158.30 (-32.49) | 59.66 (-49.01) |
| PRO-SCALE (1, 1, 1) | 50.79 | 62.39 | 39.57 | 137.10 (-41.54) | 22.89 (-80.44) |
| PRO-SCALE (2, 2, 2) | 52.12 | 63.21 | 41.58 | 154.40 (-34.16) | 39.53 (-66.21) |
| PRO-SCALE (3, 3 ,3) | 52.82 | 63.49 | 42.60 | 171.70 (-26.78) | 56.18 (-51.99) |
| **Backbone: Res50** | | | | | |
| M2F (Cheng et al., 2022) | 51.73 | 61.94 | 41.72 | 229.10 | 135.00 |
| MF (Cheng et al., 2021) | 46.50 | 57.80 | 33.00 | 181.00 (-20.99) | – (–) |
| PEM (Cavagnero et al., 2024) | 46.38 | 55.95 | 34.25 | 110.90 (-51.59) | – (–) |
| YOSO (Hu et al., 2023) | 48.40 | 58.74 | 36.87 | 114.50 (-50.02) | – (–) |
| RAP-SAM (Xu et al., 2024) | 46.90 | – | – | 123.00 (-46.31) | – (–) |
| ReMaX (Sun et al., 2023) | 53.50 | – | – | 169.00 (-26.23) | – (–) |
| PRO-SCALE (1, 1, 1) | 50.31 | 60.66 | 39.56 | 131.40 (-42.65) | 30.25 (-77.59) |
| PRO-SCALE (2, 2, 2) | 51.21 | 61.53 | 40.66 | 148.80 (-35.05) | 48.85 (-63.48) |
| PRO-SCALE (3, 3, 3) | 51.45 | 61.58 | 41.45 | 166.10 (-27.50) | 67.45 (-50.03) |

2022), we report **AP**$_p$ (Average-Precision (Lin et al., 2014)) metric for instance segmentation. This is computed on the 'thing' categories from the instance segmentation annotations. For semantic segmentation, we report **mIoU**$_p$ (mean Intersection-over-Union (Everingham et al., 2015)) by merging instance masks from the same category. Here, subscript $p$ denotes evaluation after training with panoptic segmentation annotations. For computing the GFLOPs, we use image scale of (800, 1333) for COCO dataset, and (1024, 2048) for Cityscapes dataset. All models are trained and evaluated on the *train* and *validation* split, respectively.

**Baselines.** We compare PRO-SCALE with the original M2F, along with some recently proposed efficient transformer encoders for detection (Li et al., 2023a; Lv et al., 2023; Cavagnero et al., 2024). Specifically, we replace the transformer encoder of M2F with encoders proposed in Lite-DETR (Li et al., 2023a) and RT-DETR (Lv et al., 2023). We call these Lite-M2F and RT-M2F, respectively. Note that, we opted for the "Lite-DETR H3L1-(6+1)×1" setup without its key-aware deformable attention (Li et al., 2023a). However, we modified this setup to (5+1) when implementing Lite-M2F. For completeness, we also compare with recent non-M2F-style panoptic segmentation models YOSO (Hu et al., 2023), RAP-SAM (Xu et al., 2024) and ReMaX (Sun et al., 2023).

**Architecture Details.** We focus on standard lightweight backbones Res50 (He et al., 2016), SWIN-Tiny (Liu et al., 2021), and MViT2-Tiny (Li et al., 2022b), pre-trained on ImageNet-1K (Deng et al., 2009). We use PRO-SCALE as the transformer encoder. As per Sec. 3, we use different integer values of $(p_1, p_2, p_3)$ to instantiate different depths of the stages in PRO-SCALE. We directly adopt the M2F's decoder that consists of masked attention (Cheng et al., 2022) with 9 layers in total and 100 learnable queries. Similar to M2F's round robin design, PRO-SCALE feeds $\{s_2, s_3, s_4\}$ into successive transformer decoder layers.

## 4.1 MAIN RESULTS

The analysis on the COCO dataset is presented in Tab. 1. We use M2F (Cheng et al., 2022) as reference for performance and present the following insights. *First*, PRO-SCALE is significantly computationally cheaper than baseline efficient M2F-style models such as Lite-M2F and RT-M2F. For example with SWIN-T backbone, PRO-SCALE achieves a PQ of 52.82 and with a computational cost of 171.70 GFLOPs. This PQ is 11.46 points better than that of RT-M2F, while enabling a ∼52% lighter transformer encoder. Compared to Lite-M2F, PRO-SCALE shows a better overall segmentation while decreasing approximately ∼52% of GFLOPs compared to ∼32% of Lite-M2F. Similar performance improvements can be observed over M2F and MaskFormer (Cheng et al., 2021) with Res50 backbone. *Second*, PRO-SCALE outperforms non-M2F-style models YOSO (Hu et al.,

Table 2: **Cityscapes evaluation.** PRO-SCALE (with configuration $(p_1, p_2, p_3)$) shows strong efficiency trade-off compared to the baselines on Cityscapes, *e.g.* 51.96% (SWIN-T) and 50.17% (Res50) GFLOPs reduction and little-to-no accuracy drop. Non-M2F-style models are colored in gray. † trained for 200K iterations.

| Model | Performance (↑) | | | GFLOPs (↓), (% decrement) | |
|---|---|---|---|---|---|
| | **PQ** | **IoU$_p$** | **AP$_p$** | **Total** | **Encoder** |
| **Backbone: SWIN-T** | | | | | |
| M2F (Cheng et al., 2022) | 64.00 | 80.77 | 39.26 | 537.8 0 | 281.00 |
| Lite-M2F (Li et al., 2023a) | 62.29 | 79.43 | 36.57 | 428.70 (-20.29) | 172.00 (-38.79) |
| RT-M2F (Lv et al., 2023) | 59.73 | 77.89 | 31.35 | 361.10 (-32.86) | 130.00 (-53.74) |
| PRO-SCALE (1, 1, 1) | 60.58 | 78.29 | 32.97 | 311.90 (-42.00) | 055.14 (-80.38) |
| PRO-SCALE (2, 2, 2) | 62.37 | 78.62 | 35.97 | 352.00 (-34.55) | 095.24 (-66.11) |
| PRO-SCALE (3, 3 ,3) | 63.06 | 77.94 | 37.81 | 392.10 (-27.09) | 135.00 (-51.96) |
| **Backbone: Res50** | | | | | |
| M2F (Cheng et al., 2022) | 61.86 | 76.94 | 37.35 | 524.10 | 291.00 |
| PEM (Cavagnero et al., 2024) | 61.07 | 77.62 | 34.11 | 236.60 (-51.59) | – (–) |
| YOSO (Hu et al., 2023) | 59.70 | 76.05 | 33.76 | 265.10 (-49.42) | – (–) |
| ReMaX† (Sun et al., 2023) | 65.40 | – | – | 294.70 (-43.77) | – (–) |
| PRO-SCALE (1, 1, 1) | 59.70 | 77.19 | 32.77 | 298.10 (-43.12) | 65.21 (-77.59) |
| PRO-SCALE (2, 2, 2) | 60.89 | 76.61 | 34.95 | 338.20 (-35.47) | 105.00 (-63.92) |
| PRO-SCALE (3, 3, 3) | 61.87 | 78.44 | 37.33 | 378.30 (-27.82) | 145.00 (-50.17) |

Table 3: **Component ablation.** Comparison of different cases with settings of PRO-SCALE for PQ and GFLOPs for the encoder with SWIN-T backbone on Cityscapes dataset. PRO-SCALE provides a flexible way to adjust performance and computational cost by varying its scaling factors.

| Case # | Setting | PQ (↑) | GFLOPs (↓) | FPS (↑) |
|---|---|---|---|---|
| | M2F | 64.00 | 281.00 | 5.10 |
| $C_1$ | M2F + (completely remove $s_1$) | 60.65 | 132.00 | 7.28 |
| $C_2$ | PRO-SCALE (1,1,1) + $\mathbf{s}_1$ | 61.73 | 73.49 | 6.36 |
| $C_3$ | PRO-SCALE (1,1,1) + ($\mathbf{s}_1$ via LPE) | 60.58 | 55.14 | 6.47 |
| $C_4$ | PRO-SCALE (2,2,2) + ($\mathbf{s}_1$ via LPE) | 62.37 | 95.24 | 6.07 |
| $C_5$ | PRO-SCALE (3,3,3) + ($\mathbf{s}_1$ via LPE) | 63.06 | 135.00 | 5.71 |

2023) in panoptic segmentation by at least ∼2 points. YOSO has a slightly better computational load than PRO-SCALE, while ReMaX (Sun et al., 2023) has competitive accuracy. Note that, ReMaX focuses on the training pipeline of panoptic segmentation (Sun et al., 2023), which is orthogonal to our approach to design efficient segmentation architectures. Further, ReMaX is limited by the inherent efficiency of model and becomes ineffective on larger models (see Appendix C in (Sun et al., 2023)). Similarly, in comparison with a recent universal segmentation architecture PEM (Cavagnero et al., 2024) with Res50 backbone, PRO-SCALE shows stronger results in severe GFLOPs reduction settings $(p_1, p_2, p_3 = 1,1,1)$. The analysis on Cityscapes dataset segmentation is shown in Tab. 2. Similar to COCO, PRO-SCALE is computationally most efficient compared to both M2F-style models (Lite-M2F and RT-M2F) and at-par with non-M2F-style models (YOSO and ReMaX (Sun et al., 2023)) with extremely competitive performance. For example, PRO-SCALE with SWIN-T backbone achieves 63.06 PQ *vs.* Lite-M2F's 62.29 PQ while having comparatively ∼27% less GLOPS than M2F. Compared to non-M2F-style models in Res50 backbone, PRO-SCALE demonstrates competitive performance with *no specific* panoptic segmentation design like YOSO (Hu et al., 2023), or much fewer training iterations with specific panoptic segmentation training strategies like ReMaX (Sun et al., 2023) (200K vs 90K iterations).

## 4.2 ABLATION STUDY

**Ablation of components.** We analyze the impact of PRO-SCALE with SWIN-T backbone in a low GFLOPs budget scenario in Tab. 3. We make the following observations. Simply removing $\mathbf{s}_1$ from M2F (Case $C_1$) reduced GFLOPs by 51.24% but also reduced PQ by 5.7%. Using PRO-SCALE $(p_1, p_2, p_3 = 1,1,1)$ (Case $C_2$) achieved a 73.85% reduction in GFLOPs with a 4.7%

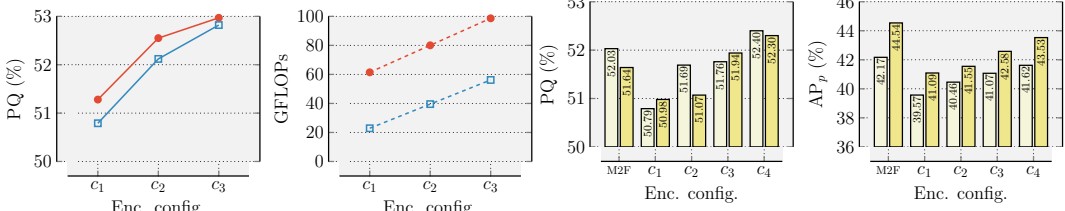

Figure 4: **Impact of LPE module.** Per-pixel embeddings produced by LPE do not significantly harm the performance but demonstrate a strong impact on the computational reduction. Here, backbone= SWIN-T, PRO-SCALE configuration: $c_1 = (1,1,1)$, $c_2 = (2,2,2)$, $c_3 = (3,3,3)$, models = w/o LPE and w/ LPE), dataset = COCO.

Figure 5: **Impact of pre-trained weights.** PRO-SCALE provides significant computational boosts, irrespective of backbone pre-trained weights. Here, backbone/dataset= SWIN-T/COCO, weights = supervised/ MoBY (Xie et al., 2021) on IN1K (Russakovsky et al., 2015), PRO-SCALE config.: $c_1 = (1,1,1)$, $c_2 = (3,1,1)$, $c_3 = (1,3,1)$, $c_4 = (1,1,3)$.

Table 4: **LPE module with Lite-M2F**. LPE can be flexibly integrated to improve the computational load of Lite-M2F (Li et al., 2023a) with minimal performance degradation. Backbone = SWIN-T.

Table 5: **Impact of different backbones.** PRO-SCALE can provide significant efficiency boosts across various backbones. Here, $(p_1, p_2, p_3) = (1, 1, 3)$, dataset = COCO.

| Dataset | Model | Performance (↑) | | | GFLOPs (% decrement) | |
|---|---|---|---|---|---|---|
| | | PQ | mIOU$_p$ | AP$_p$ | Total | Encoder |
| COCO | Lite-M2F | 52.70 | 63.52 | 42.26 | 188.00 | 79.78 |
| | + LPE | 52.32 | 63.08 | 41.10 | 154.60 (-17.77) | 43.92 (-44.95) |
| Cityscapes | Lite-M2F | 62.29 | 79.43 | 36.57 | 428.70 | 172.00 |
| | + LPE | 61.35 | 79.05 | 35.91 | 351.40 (-18.03) | 94.69 (-44.95) |

| Backbone type | Performance (↑) | | | GFLOPs (↓), (% decrement) | |
|---|---|---|---|---|---|
| | PQ | mIOU$_p$ | AP$_p$ | Total | Encoder |
| SWIN-T | 52.03 | 62.49 | 42.17 | 234.50 | 117.00 |
| | 52.40 | 62.66 | 41.62 | 164.70 (-29.77) | 54.77 (-53.19) |
| Res50 | 51.73 | 61.94 | 41.72 | 229.10 | 135.00 |
| | 51.35 | 61.53 | 40.82 | 158.60 (-30.77) | 59.44 (-55.97) |
| MViT-T | 54.11 | 64.39 | 44.54 | 244.60 | 130.00 |
| | 53.70 | 64.17 | 43.53 | 174.10 (-28.82) | 54.77 (-57.87) |

PQ reduction. When $\mathbf{s}_1$ is added back via LPE to $(p_1, p_2, p_3 = 1,1,1)$ (Case $C_3$), it results in a stronger performance-efficiency trade-off than $C_1$. This performance further improves when different configurations of PRO-SCALE are used in case $C_4$ and $C_5$.

**Impact of LPE module.** Our LPE module is aimed to drastically decrease the computational overhead without impairing the segmentation performance in PRO-SCALE. To analyze this, we compare LPE with the M2F's convolutional layer based embedding layer module in Fig. 4. Here, we can observe that compared to this convolutional unit with learnable parameters, LPE is extremely effective. For example, for PRO-SCALE configuration $c_3 = (3,3,3)$, LPE reduces the computations by ~40% with performance degradation of 0.12 PQ points. Similarly, $c_2 = (1,1,1)$ and $c_2 = (2,2,2)$ provide similar computational benefits with 0.43 PQ points degradation. This demonstrates the favorable performance-computation trade-off LPE provides in PRO-SCALE. To further demonstrate the potency of our proposed LPE module, we incorporate LPE in Lite-M2F and analyze the results in Tab. 4. We can observe that LPE shows similar impact on Lite-M2F: ~45% less transformer encoder GLOPS with only ~0.5-1 PQ point trade-off.

**Variations in encoder configuration.** Here, we analyze the impact of PRO-SCALE configuration on model performance in Tab. 6. We can make some key observations. *First*, we observe that PRO-SCALE shows stronger performance if any $p_i > 1$. This is trivial as more computations are allowed for the corresponding layers. *Second*, increasing $p_3$ and $p_2$ have different performance-efficiency trade-off: for COCO, $p_2$ *vs.* $(p_2 + 2)$ results in only increment of $\sim 6.5$ GFLOPs with $\sim 1.0\%$ PQ improvement whereas $p_3$ *vs.* $(p_3 + 2)$ results in a higher computational load of $\sim 32$ GFLOPs increase with $\sim 1.6\%$ PQ performance improvement. We make similar observations for the Cityscapes dataset for $(p_1 + 2)$ *vs.* $(p_2 + 2)$.

**Variations in backbone architectures.** We analyze the performance of PRO-SCALE with different backbones in Tab. 5. Specifically, we observe the impacts of SWIN-T, Res50, and MViT-T in the segmentation performance. PRO-SCALE is versatile in providing strong performance-efficiency across diverse backbone types. We also present results with larger backbones namely ResNet101,

Table 6: **Analysis of configurations** $(p_1, p_2, p_3)$. Increments in different $p_i$ shows different performance-efficiency trade-off.

| $p_1, p_2, p_3$ | Performance (↑) | | | GFLOPs (↓), (% decrement) | |
|---|---|---|---|---|---|
| | PQ | mIOU$_p$ | AP$_p$ | Total | Encoder |
| **Dataset**: Cityscapes, **Backbone**: SWIN-T | | | | | |
| original | 64.00 | 80.77 | 39.26 | 537.80 | 281.00 |
| (1, 1, 1) | 60.58 | 78.29 | 32.97 | 311.90 (-42.00) | 055.14 (-80.38) |
| (3, 1, 1) | 61.38 | 78.58 | 33.75 | 314.70 (-41.48) | 057.93 (-79.38) |
| (1, 3, 1) | 61.63 | 78.67 | 35.26 | 326.30 (-39.33) | 069.62 (-75.22) |
| (1, 1, 3) | 62.32 | 79.49 | 36.97 | 374.80 (-30.31) | 118.00 (-58.01) |
| **Dataset**: COCO, **Backbone**: SWIN-T | | | | | |
| original | 52.03 | 62.49 | 42.17 | 234.50 | 117.00 |
| (1, 1, 1) | 50.79 | 62.39 | 39.57 | 137.10 (-41.54) | 22.89 (-80.44) |
| (3, 1, 1) | 51.69 | 62.81 | 40.46 | 138.70 (-40.85) | 26.87 (-77.03) |
| (1, 3, 1) | 51.76 | 63.23 | 41.07 | 143.80 (-38.68) | 32.29 (-72.40) |
| (1, 1, 3) | 52.40 | 62.66 | 41.62 | 164.70 (-29.77) | 54.77 (-53.19) |

Table 7: **Impact of large backbones.** PRO-SCALE can provide significant efficiency boosts across various backbones. Here, $(p_1, p_2, p_3) = (3, 3, 3)$, dataset = Cityscapes.

| Backbone type | Performance (↑) | | | GFLOPs (↓), (% decrement) | |
|---|---|---|---|---|---|
| | PQ | mIOU$_p$ | AP$_p$ | Total | Encoder |
| SWIN-S | 64.80 | 81.80 | 40.70 | 724.30 | 281.00 |
| | 64.41 | 80.19 | 39.90 | 578.50 (-20.13) | 135.00 (-51.96) |
| SWIN-B | 66.10 | 82.70 | 42.80 | 1051.20 | 283.00 |
| | 65.27 | 81.97 | 41.12 | 905.40 (-13.87) | 137.00 (-51.59) |
| SWIN-L | 66.60 | 82.90 | 43.60 | 1949.70 | 287.00 |
| | 65.93 | 82.77 | 41.80 | 1803.90 (-7.48) | 141 (-50.87) |
| Res101 | 62.40 | 78.60 | 37.70 | 679.70 | 291.00 |
| | 61.28 | 76.50 | 36.49 | 533.90 (-21.45) | 145.00 (-50.17) |

SWIN-Small, SWIN-Base, and SWIN-Large on Cityscapes dataset in Fig. 7. PRO-SCALE reduces GFLOPs in all cases while providing a strong accuracy-efficiency trade-off. The PRO-SCALE configuration is $(p_1, p_2, p_3) = (3, 3, 3)$. Note that, we can also employ a different PRO-SCALE configuration $(p_1, p_2, p_3)$ to further reduce the computations.

**Variations in backbone pre-trained weights.** We analyze the performance of PRO-SCALE with different pre-training strategies of the backbones in Fig. 5. Specifically, we initialize SWIN-T with *supervised learning* (SL) and *self-supervised learning* (SSL) based ImageNet-1K weights. For SSL pre-training, we employ MoBY (Xie et al., 2021). We can see that PRO-SCALE can gain performance robustness of SSL weights while providing the same trade-offs as SL weights. For example, when $c_1 = (p_1, p_2, p_3) = (1, 1, 1)$, MoBY shows an improvement of ∼1.6% AP$_p$ with the same efficiency gains. On average, integrating PRO-SCALE with the MoBY pre-trained backbone results in better (overall) performance compared to using SL backbone weights, especially in instance segmentation (Fig. 5, *right*).

**FPS comparison.** Tab. 9 compares M2F and PRO-SCALE against state-of-the-art efficient encoder for universal segmentation method PEM (Cavagnero et al., 2024). While M2F achieves the highest PQ (51.73) and lowest FPS (4.91) and PEM provides the highest FPS (7.31) but lower PQ (46.38), PRO-SCALE offers a competitive PQ (51.35) with a balanced FPS (6.25).

**Impact beyond universal segmentation.** We analyze PRO-SCALE on tasks beyond universal segmentation to test its effectiveness. In particular, we add PRO-SCALE to DINO (Zhang et al., 2022) for the task of object detection, MaskDINO (Li et al., 2022a) for the joint task of instance segmentation-detection, and FCCLIP (Yu et al., 2023) and MaskCLIP (Zheng Ding, 2023) for the task of open-vocabulary segmentation. Note that we can use a different PRO-SCALE configuration to further reduce the computations in all cases.

- **Impact on DINO for object detection.** As shown in Tab. 8, PRO-SCALE (integrated with DINO) stands out with the highest AP (49.4), showcasing exceptional performance compared to other models. It also has the lowest encoder GFLOPs (56.18), making it more computationally efficient than other methods. We also show segmentation PQ for comprehensive overview.

- **Impact on two open-vocab universal segmentation frameworks.** We used recent state-of-the-art open-vocabulary image segmentation methods MackCLIP (Zheng Ding, 2023) and FCCLIP (Yu et al., 2023). Our method PRO-SCALE can easily work with these frameworks and reduce the computations while having better performance as shown in Tab. 10. We followed the exact training and testing protocols of respective methods and trained our model with COCO. We perform evaluation on COCO val set and cross-evaluation on ADE20K (Zhou et al., 2017) val set.

- **Impact on MaskDINO for multi-task prediction.** We integrated and trained MaskDINO with PRO-SCALE, and the results are shown in Tab. 11. In particular, we trained the model on COCO panoptic annotations with the exact same training settings as MaskDINO for 50 epochs and evaluated the model for segmentation and object detection. Clearly, PRO-SCALE effectively reduces the computational requirements of MaskDINO while maintaining overall performance.

Table 8: PRO-SCALE **for object detection**. PRO-SCALE can easily work with DETR (Carion et al., 2020) based object detection models and reduce the computations while having better performance. The PRO-SCALE configuration is $(p_1, p_2, p_3, p_4)$ = (3,3,3,3), backbone is Res50, training dataset is COCO, and trained for 12 epochs. The segmentation results are with SWIN-T backbone. LPE module is not required for object detection.

| Model | Performance (↑) | | GFLOPs (↓) | |
|---|---|---|---|---|
| | AP | PQ | Total | Encoder |
| DINO (Zhang et al., 2022) | 49.00 | N/A | N/A | N/A |
| Lite-DETR (Lv et al., 2023) | 49.10 | 52.70 | 188.00 | 79.78 |
| RT-DETR (Li et al., 2023a) | 49.20 | 41.36 | 158.30 | 59.66 |
| PRO-SCALE | 49.40 | 52.82 | 171.70 | 56.18 |

Table 9: PRO-SCALE **FPS comparison**. PRO-SCALE strikes a balance with a high PQ score, and good FPS, offering a good trade-off between quality and speed. The PRO-SCALE configuration is $(p_1, p_2, p_3)$ = (3,3,3), backbone is Res50, and dataset is COCO.

| Model | PQ (↑) | FPS (↑) |
|---|---|---|
| M2F (Cheng et al., 2022) | 51.73 | 4.91 |
| PEM (Cavagnero et al., 2024) | 46.38 | 7.31 |
| PRO-SCALE | 51.45 | 6.25 |

Table 10: PRO-SCALE **for open-vocabulary segmentation**. PRO-SCALE can easily work with M2F based open-vocab universal segmentation models (Zheng Ding, 2023; Yu et al., 2023) and reduce the computations while having better PQ performance. The PRO-SCALE configuration is $(p_1, p_2, p_3)$ = (3,3,3), backbone is Res50, and training dataset is COCO.

| Model | Performance (↑) | | GFLOPs (↓) | |
|---|---|---|---|---|
| | COCO | ADE20K | Total | Encoder |
| MaskCLIP (Zheng Ding, 2023) | 29.98 | 15.12 | 549.20 | 135.00 |
| + PRO-SCALE | 34.53 | 16.53 | 486.10 | 67.45 |
| FCCLIP (Yu et al., 2023) | 54.40 | 26.80 | 2119.60 | 287.00 |
| + PRO-SCALE | 55.81 | 25.27 | 1973.80 | 141.00 |

Table 11: PRO-SCALE **for MaskDINO**. PRO-SCALE can easily work with MaskDINO models (Li et al., 2022a) and reduce the computations while having better Mask and Box AP performance. The PRO-SCALE configuration is $(p_1, p_2, p_3)$ = (3,3,3), backbone is Res50, and training dataset is COCO.

| Model | Performance (↑) | | GFLOPs (↓) | |
|---|---|---|---|---|
| | Mask AP | Box AP | Total | Encoder |
| MaskDINO | 46.00 | 50.50 | 302.10 | 204.00 |
| + PRO-SCALE | 45.66 | 50.83 | 212.40 | 108.00 |

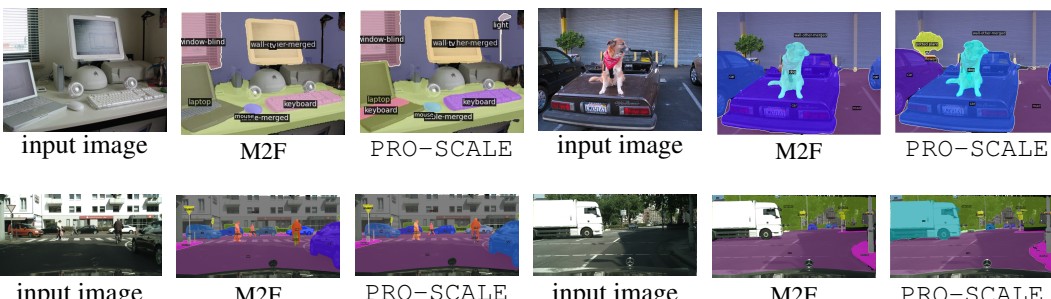

| input image | M2F | PRO-SCALE | input image | M2F | PRO-SCALE |

| input image | M2F | PRO-SCALE | input image | M2F | PRO-SCALE |

Figure 6: **Qualitative visualizations.** We visualize examples of predicted panotic maps from M2F and PRO-SCALE. Even with 52% encoder GFLOPs reduction, PRO-SCALE shows better quality panoptic maps (backbone = SWIN-T, PRO-SCALE config. = (3, 3, 3)). *Zoom-in for best view.*

**Qualitative analysis.** We show some examples of predicted panoptic maps in Fig. 6 with SWIN-T backbone. We set PRO-SCALE configuration to (3, 3, 3). Compared to M2F (Cheng et al., 2022), PRO-SCALE consistently shows strong performance with drastically fewer GLOPs in everyday scenes (*top* row on COCO) as well as complex driving scenes (*bottom* row on Cityscapes).

## 5 CONCLUSION

In this paper, we propose an efficient transformer encoder PRO-SCALE for the Mask2Former universal segmentation framework. It reduces the computational load by a large margin with minimal degradation in performance. The core principle of PRO-SCALE is to progressively expand the length of the tokens with the layers of the encoder. Further, a light per-pixel embedding module is introduced to alleviate the computational overhead arising from creating per-pixel embeddings without much sacrifice in performance. Our extensive experiments demonstrate that PRO-SCALE is significantly lighter than prior prominent methods while maintaining competitive universal segmentation performance.

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

## A  EXPERIMENT DETAILS

**Datasets.**    We examine PRO-SCALE with two image segmentation datasets: COCO (Lin et al., 2014) (80 "things", 53 "stuff" categories) and Cityscapes (Cordts et al., 2016) (8 "things", 11 "stuff" categories). COCO has 118K training and 5K validation images. Cityscapes has 2975 training and 500 validation images.

**Training details.**    All our training details and underlying strategy follow (Cheng et al., 2022), including for our baselines. We use Detectron2 (Wu et al., 2019) and PyTorch (Paszke et al., 2019) for implementation. We employ the AdamW optimizer (Loshchilov & Hutter, 2017) with a step learning rate schedule. The initial learning rate is 0.0001. The backbone learning rate is multiplied with 0.1 and its weight decay is set as 0.05. We decay the learning rate at 0.9 and 0.95 fractions of the total training steps by a factor of 10. For COCO, our models are trained for 50 epochs. For Cityscapes, we use 90k iterations. Batch size is set as 16. For data augmentation and calculating GFLOPs, we follow the strategies exactly as M2F. We report the average GFLOPs for all cases. We use distributed training with 8 A6000 GPUs.

**Loss functions.**    We use the exact same loss functions and weights as (Cheng et al., 2022). In particular, we use the binary cross-entropy loss and the dice loss (Milletari et al., 2016) for our mask loss. Both loss functions use a weight of 5.0. The final loss is a combination of mask loss and classification loss (cross-entropy loss). We set the classification loss weight as 2.0 for all classes, except 0.1 for the 'no object' class. Finally, we apply the identical post-processing methodology as described in (Cheng et al., 2022) to obtain the desired output formats for panoptic, semantic, and instance segmentation predictions.

## B  ADDITIONAL RESULTS

**LPE pooling analysis.**    We analyzed the LPE module with 'average pooling' in place of 'max-pooling' on the Cityscapes dataset with Res50 backbone and $(p_1, p_2, p_3) = (3,3,3)$. We observed that the performance of the LPE module performs better with max-pooling (61.87% PQ) than average pooling (61.47% PQ).

**Token redundancy visualization.**    We provide an explicit qualitative visualization that proves the token redundancy in early stages of the transformer encoder in Fig. 9. This analysis is on 100 randomly chosen images from COCO val set. In this figure, the $x$-axis represents the distance of neighborhood tokens from the candidate token (along the token dimension) and the $y$-axis represents the (average across the number of images) cosine similarity between the two. It can be observed that in larger scale tokens, the token similarity is higher than smaller token as the neighborhood distance increases. This clearly suggests that the information redundancy of larger scale tokens is comparatively higher than smaller scale tokens.

**Efficiency vs Performance overview.**    Fig. 10 illustrates the trade-off between efficiency (GFLOPS on the x-axis) and performance (PQ on the y-axis) for different configurations of $(p_1, p_2, p_3)$ on the COCO dataset, with each point representing a specific configuration. The red dashed line indicates the original model's performance (PQ = 52.03), highlighting how the configurations compare in terms of maintaining performance while reducing computational costs.

**Additional qualitative visualization.**    We show some examples of predicted panoptic maps in Fig. 11 with Res50 backbone on the COCO dataset. We set PRO-SCALE configuration to (3, 3, 3). Compared to Lite-M2F (Li et al., 2023a) and PEM (Cavagnero et al., 2024), PRO-SCALE consistently shows strong performance with drastically fewer GLOPs.

**FPS analysis.**    The following analysis is with A6000 GPUs. We analyzed the gain in speed when PRO-SCALE is integrate with different backbones in Tab. 16. Further, Tab. 15 provides an overview of FPS with different baselines. PRO-SCALE provides good trade-off between performance and speed in different settings.

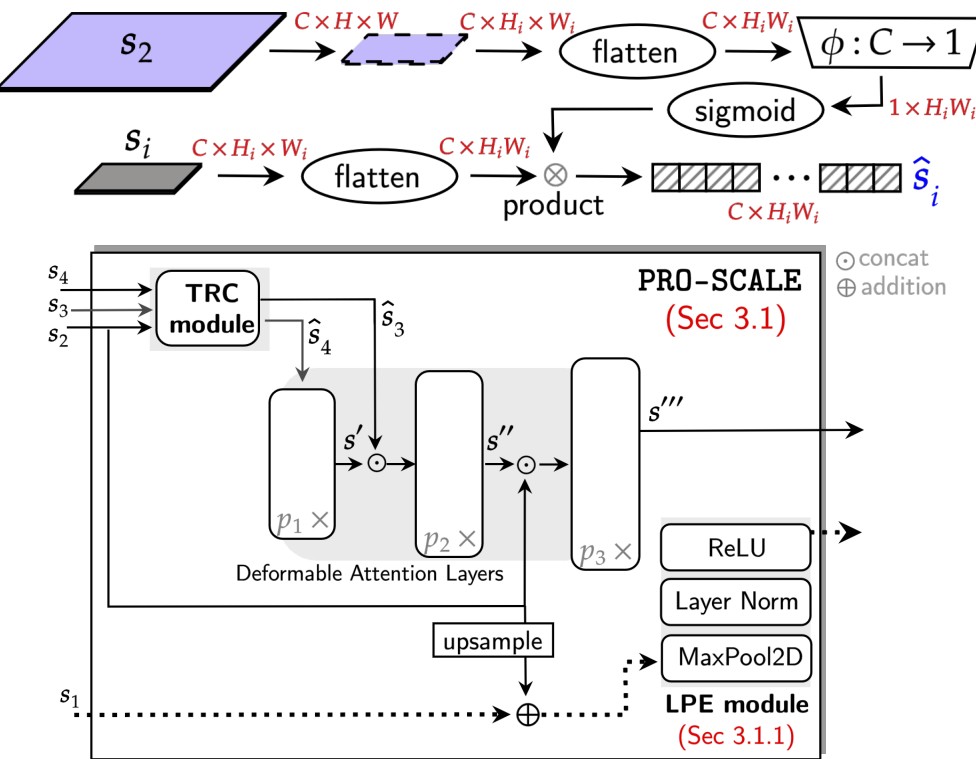

Figure 7: **Token Re-Calibration (TRC) module.** This unit first interpolates $s_2$ to the size of the target smaller-scale feature $s_i$. Then, the spatially flattened and resized $s_2$ is passed through a channel reduction layer $\phi$ (followed by sigmoid function) to produce an attention map. This map is imposed on the spatially flattened $s_i$ to create $\widehat{s}_i$.

## C  PRO-SCALE ADDENDUM: TOKEN RE-CALIBRATION (TRC) MODULE

**Intuition.** Skipping the propagation of $s_2$ in the initial layers may lead to errors in predicted panoptic maps due to missing information from this feature scale. To avoid such map errors caused by imperfect token representations, we propose to calibrate the $s_3$ and $s_4$ using $s_2$ in our TRC module. This module aims to enhance small-scale features $s_i \in \{s_3, s_4\}$ using large-scale feature $s_2$ without increasing the computations in PRO-SCALE transformer layers.

**TRC structure.** To utilize strengths of $s_2$ in smaller scales, we employ contrastive attention. In particular, we propose to enrich the tokens of $s_i$ by projecting $s_2$ on $s_i$ *via* an attention map. This attention map is obtained by using a channel reduction layer $\phi$ followed by a sigmoid function. As illustrated in Fig. 7, the final tokens $\widehat{s}_i$ are obtained as ($\otimes$ represents elementwise multiplication):

$$\widehat{\mathbf{s}}_i = \mathbf{s}_i \otimes \texttt{sigmoid}(\epsilon\phi(\mathbf{s}_2)) \tag{4}$$

With above, Eq. 2, 3 translate to $\mathbf{P}_1 = \mathcal{C}(\widehat{\mathbf{s}}_4)$ and $\mathbf{P}_2 = \mathcal{C}(\mathbf{s}', \widehat{\mathbf{s}}_3)$. We choose $\phi(\cdot)$ to be a linear layer and set temperature $\epsilon = 0.1$, keeping computations negligible.

**Impact of TRC module.** Our TRC module is designed to incorporate the benefits of large scale feature $s_2$ without increasing computational overhead in PRO-SCALE. In Tab. 12, we can see that the TRC module consistently improves the performance of the proposed framework demonstrating the efficacy of this unit. Moreover, this unit only costs 0.006 GFLOPs for the COCO dataset. We also analyze the impact of $\epsilon$ on the TRC module in Tab. 13 and find that it overall improves the performance of segmentation with best performance at $\epsilon = 0.1$. Finally, we provide visualizations of impact of TRC module in two examples shown in Fig. 8. It can be observed that the TRC module introduces stronger focus on parts of images that need to be segmented by the model.

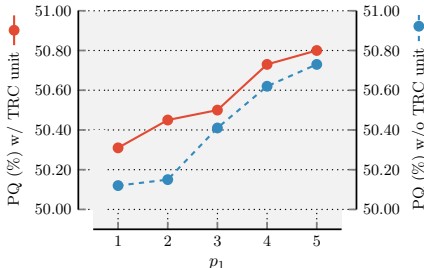

Table 12: **Impact of TRC module.** Use of $\mathbf{s}_2$ *via* TRC to enhance $\{\mathbf{s}_3, \mathbf{s}_4\}$ improves the overall performance with light computational load. Here, backbone = Res50, dataset = COCO, config.=$(p_1, 1, 1)$.

Table 13: **Impact of $\epsilon$ on TRC.** Temperature scaling $\epsilon$ of 0.1 shows the best performance. Here, backbone = SWIN-T, dataset = Cityscapes.

| $\epsilon$ | PQ(%) |
|---|---|
| 0.01 | 60.05 |
| 0.10 | **60.58** |
| 1.00 | 60.32 |
| 5.00 | 60.41 |
| 10.00 | 60.34 |

input image     before TRC module     after TRC module

input image     before TRC module     after TRC module

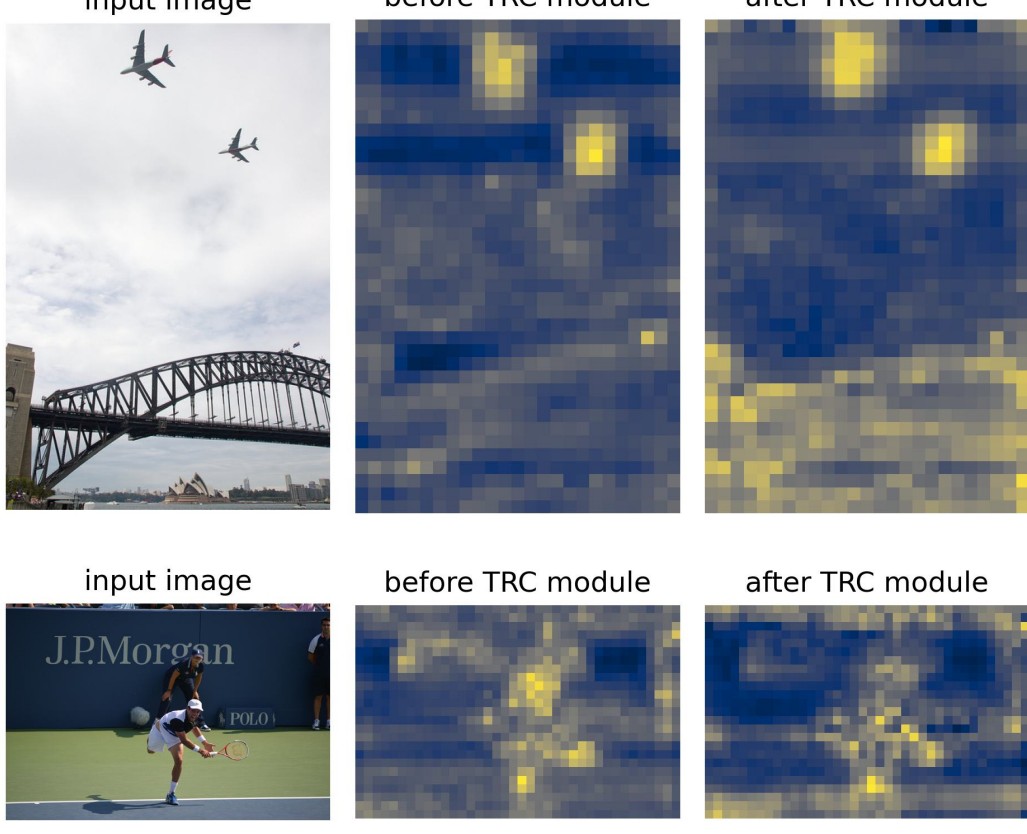

Figure 8: **Effect of TRC module.** It can be observed that the TRC module introduces stronger focus on parts of images that need to be segmented by the model.

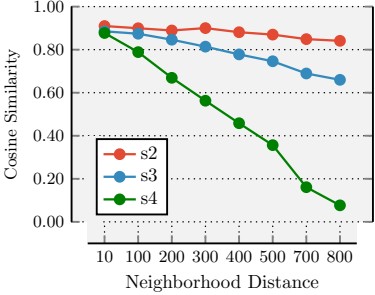

Figure 9: **Token redundancy visualization.**

Table 14: **LPE pooling analysis**. We observe that max-pooling provides better results with PRO-SCALE than average pooling. The PRO-SCALE configuration is $(p_1, p_2, p_3) = (3,3,3)$, backbone is Res50, training dataset is Cityscapes.

| Pooling | PQ ($\uparrow$) |
|---|---|
| avg-pooling | 61.47 |
| max-pooling | 61.87 |

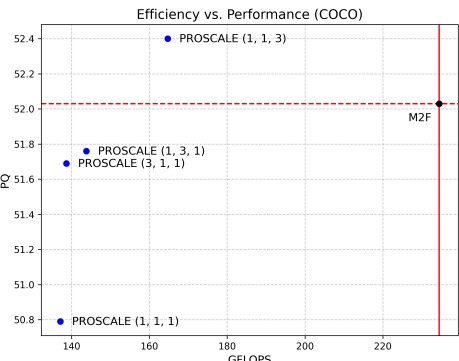

Figure 10: Efficiency vs. Performance for different PRO-SCALE $(p_1, p_2, p_3)$ configurations on COCO. The red lines indicate M2F's PQ and GFLOPs for reference.

Table 15: PRO-SCALE **FPS comparison with baselines**. PRO-SCALE strikes a balance with a high PQ score, and good FPS, offering a good trade-off between quality and speed. Backbone is SWIN-T, and dataset is Cityscapes.

| Model | PQ ($\uparrow$) | FPS ($\uparrow$) |
|---|---|---|
| Lite-M2F (Cheng et al., 2022) | 62.29 | 6.01 |
| RT-M2F (Cavagnero et al., 2024) | 59.73 | 6.59 |
| PRO-SCALE (1,1,1) | 60.58 | 6.47 |
| PRO-SCALE (2,2,2) | 62.37 | 6.07 |
| PRO-SCALE (3,3,3) | 63.06 | 5.71 |

Table 16: PRO-SCALE **FPS comparison on different backbones**. PRO-SCALE strikes a balance with a high PQ score, and good FPS, offering a good trade-off between quality and speed. Here, $(p_1, p_2, p_3) = (1, 1, 3)$, dataset = COCO.

| Backbone type | Performance ($\uparrow$) | | | FPS ($\uparrow$) |
|---|---|---|---|---|
| | PQ | mIOU$_p$ | AP$_p$ | |
| SWIN-T | 52.03 | 62.49 | 42.17 | 5.09 |
| | 52.40 | 62.66 | 41.62 | 5.28 |
| Res50 | 51.73 | 61.94 | 41.72 | 2.10 |
| | 51.35 | 61.53 | 40.82 | 5.71 |
| MViT-T | 54.11 | 64.39 | 44.54 | 2.08 |
| | 53.70 | 64.17 | 43.53 | 4.85 |

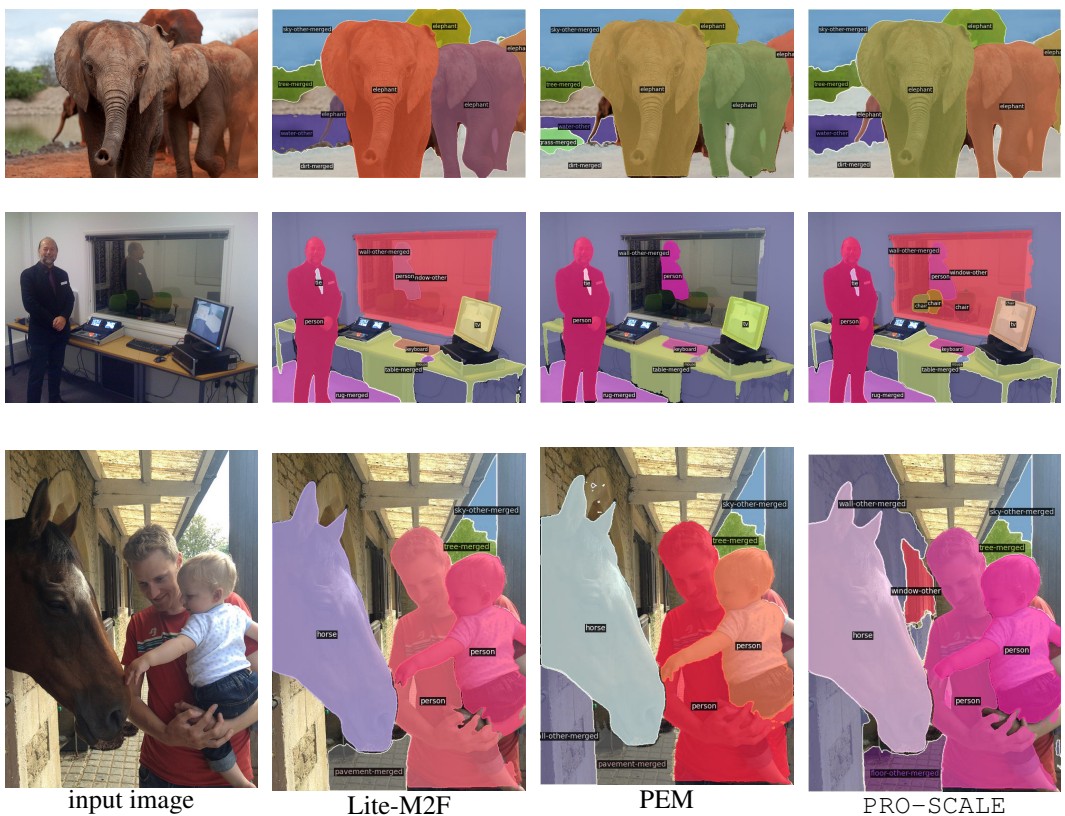

input image      Lite-M2F      PEM      PRO-SCALE

Figure 11: **Qualitative visualizations.** We visualize few examples of predicted panotic maps from Lite-M2F (Li et al., 2023a) and PEM (Cavagnero et al., 2024) and PRO-SCALE. Even with 52% transformer encoder GFLOPs reduction, PRO-SCALE shows better quality panoptic maps. Here, backbone = Res50, PRO-SCALE configuration = (3, 3, 3). *Zoom-in for best view*.

