# OpenReview forum: "Progressive Token Length Scaling in Transformer Encoders for Efficient Universal Segmentation"
_ICLR.cc/2025/Conference — ICLR 2025 Poster_

### Official Review · Reviewer_zhiy · 2024-10-30

**Soundness:** 2
**Presentation:** 2
**Contribution:** 2
**Rating:** 5
**Confidence:** 4

**Summary:**

This paper presents *PRO-SCALE*, a method to improve the efficiency of the “transformer encoder” component of the state-of-the-art Mask2Former model for universal image segmentation. In Mask2Former, this transformer encoder (also called ‘pixel decoder’ [a]) takes the tokens of multi-resolution features from the backbone, applies multiple layers of deformable attention to all tokens, and then feeds them to the segmentation decoder. In contrast, PRO-SCALE applies the first deformable attention layers only to a small number of low-resolution feature tokens, and then gradually adds higher-resolution features in subsequent layers. By doing so, the number of operations is lower than in the original transformer encoder. Additionally, to efficiently generate the highest-resolution features that the decoder needs, PRO-SCALE contains a *Light Pixel Embedding* (LPE) module that applies max-pooling to high-resolution features from the backbone. With experiments on multiple datasets and with multiple backbones, PRO-SCALE is shown to reduce the total number of FLOPs by up to 27%, while keeping the segmentation quality (PQ, mIoU, AP) roughly the same as for Mask2Former. Additional experiments show that PRO-SCALE is also effective in combination with open-vocabulary and object detection models.

[a] Cheng et al., “Masked-attention Mask Transformer for Universal Image Segmentation”, CVPR 2022.

**Strengths:**

1.	The proposed method is simple but effective. It is simple because there are only two relatively small changes with respect to the original transformer encoder of Mask2Former: (a) the first deformable attention layers are applied to only a subset of the feature tokens, and (b) the model applies max-pooling on the highest-resolution features. By doing so, PRO-SCALE can reduce the total number of FLOPs of Mask2Former by up to 27%, without causing a drop in Panoptic Quality (PQ), mean IoU or Average Precision (AP). Although the adjustments to Mask2Former are minimal and the technical innovation of PRO-SCALE is limited, I believe it is valuable because the paper critically examines an inefficient component of a frequently used model, and finds that it can be made considerably more efficient while keeping the performance roughly the same, with only minor changes to the design.
2.	Through experiments, PRO-SCALE is not only compared to Mask2Former with its default transformer encoder, but also to existing efficient versions of the transformer encoder that were proposed for object detection method DETR, i.e., Lite-M2F and RT-M2F in Tab. 1 and Tab. 2. The results of these comparisons show that PRO-SCALE also achieves a better efficiency-accuracy balance than these existing methods. This demonstrates the value of PRO-SCALE compared to these existing methods.
3.	The paper contains many ablations and additional experiments (Tab. 3 to Tab. 7), which properly show the impact of different design choices and demonstrate the effectiveness of PRO-SCALE in different experimental settings, e.g., with different backbones or pre-training.
4.	Through experiments, the paper shows that PRO-SCALE is not only effective on Mask2Former, but also on other models for other tasks. The results in Tab. 8, Tab. 10 and Tab. 11 show that PRO-SCALE achieves similar FLOPs reductions in the encoder when combined with object detection model DETR, two open-vocabulary segmentation models, and instance segmentation model Mask-DINO, while obtaining similar or better segmentation performance. This shows that the proposed PRO-SCALE method is more generally applicable than just on Mask2Former.

**Weaknesses:**

1.	The paper does not clearly explain how the original Mask2Former model generates the high-resolution *per-pixel embedding map* ${\mathcal{E}_{emb}}$, and why this is less efficient than how PRO-SCALE does it with LPE. L242-L243 states that $\mathbf{s}_1$ serves the purpose of creating the per-pixel embedding map, and that it uses a convolutional layer, but there is no clear description or depiction of the exact manner in which this is done. As a result, it is also not clear why the newly proposed LPE module is more efficient.
2.	Related to this, some details are missing about the exact operation of the LPE method. Concretely, what is the stride of the MaxPool2D operation? Is it the same as the kernel size? If the stride is >1, then what is the impact of changing the stride (and therefore the resolution of ${\mathcal{E}_{emb}}$) on the results, both qualitatively and quantitatively? This information is currently not available.
3.	Sec. 3.1 (L233-L236) briefly mentions that PRO-SCALE additionally uses a so-called *token recalibration* operation, which enriches small-scale features with high-scale features to further enhance the segmentation accuracy. However, despite being part of the PRO-SCALE method, this operation is not visualized in Fig. 3, and not explained properly in Sec. 3.1. As a result, (a) Fig. 3 makes it seem like this operation does not exist at all, and (b) it is unclear from the main paper how a part of the method works. In other words, this *token recalibration* operation should be explained and visualized in the main paper.
4.	Related to the previous point, the efficiency of the *token recalibration* operation is not evaluated, neither in the main paper nor in the appendix. As a result, it is not clear if it is actually efficient.
5.	For most experiments, the paper reports the FLOPs but not the latency/FPS/throughput of the model. The FPS is only reported for the overall model in Tab. 9. As a result, for most configurations, it is not clear how/if the reduction in FLOPs translates to a speedup when the model is run on a GPU. The results of the paper would be stronger if the FPS/latency of the model was also reported for other experiments, i.e., at least for the main results in Tab. 1 and Tab. 2, and the ablations in Tab. 3, Tab. 5 and Fig. 4, but ideally even more.
6.	PRO-SCALE does not yield a significant efficiency improvement when used in combination with large backbones, e.g., Swin-L (see Tab. 7). Of course, this is expected, as the Swin-L backbone accounts for a much larger portion of the overall number of FLOPs than Swin-T, but it is still a weakness because PRO-SCALE’s value in larger models is limited.
7.	The abstract contains a misleading statement. L023 states that PRO-SCALE can achieve a ~52% GFLOPs reduction with no drop in performance on the COCO dataset. This statement implies that, compared to the original Mask2Fomer, the overall PRO-SCALE model requires ~52% fewer GFLOPs. However, Tab. 1 shows that this 52% GFLOPs reduction concerns the encoder only, and that the overall GFLOPs reduction of the model is ~27%. The statement in the abstract should be altered by either specifying that the 52% reduction concerns the encoder, or changing the number from 52% to 27%.
8.	Tab. 8, Tab. 10 and Tab. 11 only contain GFLOPs results for the encoder, not for the entire model. As a result, it is not clear what the actual overall improvement is of PRO-SCALE.
9.	L323 & L358 state that ReMaX is limited by the inherent efficiency of the model, but the efficiency for ReMaX is not provided in Tab. 1 or Tab. 2. In other words, this claim is not substantiated.
10.	L454-L455 states that, on average, the MoBY pre-trained backbone causes lower performance degradation than SL pre-trained weights, especially for instance segmentation. However, per Fig. 5 (right), compared to the Mask2Former baseline, the average drop for MoBY over all 4 settings is -2.35 AP, while it is -1.49 for supervised learning. Therefore, this statement is incorrect. This statement is not very important for the main message of the paper, so it doesn’t impact the value of the proposed method, but it should be altered nevertheless.

Some minor points, which do not significantly affect my overall rating:

11.	The text contains several mistakes/errors. Some examples:

    a.	L039 – “framework exhibit exceptional performance” => “framework exhibits exceptional performance”

    b.	L136 – “making Mask2Former universal segmentation model” => “making the Mask2Former universal segmentation model”

    c.	L153 – “map and class prediction heads” => “mask and class prediction heads”

    d.	L358 – Not clear what is meant by “becomes ineffective efficient”

    e.	L377 – "effieiciency" => "efficiency"

    f.	L376-L377 – “strong … than” => “stronger … than”

    g.	L436 – “$p_2$ vs. ($p_2$ + 3)” should be “$p_2$ vs. ($p_2$ + 2)”, as the baseline is $p_2 = 1$ and the comparison is “1 vs. 3”.

    h.	L437 – Likewise: “$p_3$ vs. ($p_3$ + 3)” => “$p_3$ vs. ($p_3$ + 2)”.

    i.	L503 – The caption of Tab. 11 states that PRO-SCALE achieves a better PQ, but the PQ is not reported in Tab. 11.

    j.	L534-L535 – “for Mask2Former universal segmentation framework” => “for the Mask2Former universal segmentation framework”

12.	In Tab. 3, the experimental setting is not explicitly indicated. The numbers correspond with Swin-T on Cityscapes, but this is not explicitly mentioned in the caption. This should be mentioned, so that the table can be understood without having to check other tables.

13.	The paper uses an inconsistent number of decimals for results on the same metrics. For instance, in Tab. 3: 132 GFLOPs for $C_1$ and 73.49 for $C_2$. It would be better if the number of decimals was consistent.

**Questions:**

I would like to ask the authors to address my concerns as formulated in the “weaknesses” section, to answer the questions posed there, and to revise the manuscript accordingly.

One additional question:

1.	How does the presence of redundancy in the feature tokens relate to the presented PRO-SCALE method? Appendix B shows that higher-resolution features have a higher cosine similarity, but it is not clear to me how this observation motivates the PRO-SCALE design, where fewer deformable attention layers are applied to high-resolution features. Could the authors clarify this relation?

---

**Update after author discussion.** After reading the different reviews, the authors' response, and the revised manuscript, and asking some follow-up questions, I decide to keep my original rating.

While I still believe that the idea of improving the efficiency of the decoder of Mask2Former can be valuable, I mainly have concerns about the actual efficiency of the proposed method. Importantly, the newly provided *prediction speed* results show that the actual efficiency improvement obtained by PRO-SCALE is limited. While PRO-SCALE obtains high FLOPs reductions, the impact on the prediction speed in terms of *frames per second* (FPS) in considerably lower. As a result, PRO-SCALE does not obtain a better *prediction speed vs. segmentation performance* balance than existing method Lite-DETR/Lite-M2F. Furthermore, it turns out that the impact of the LPE module on the prediction speed is almost negligible, while it also causes a segmentation performance drop.

Because the main purpose of the paper is to improve efficiency while keeping the segmentation performance as high as possible, I believe the value of the paper is considerably limited when (a) the proposed method is not shown to obtain a better *prediction speed vs. segmentation performance* balance than a highly related existing method (Lite-M2F/Lite-DETR), and (b) the actual efficiency improvement (in terms of FPS) of one of the main contributions (LPE module) is almost negligible while causing a drop in segmentation performance.

In their response, the authors mention that other benefits of reduced FLOPs are (i) improved energy efficiency because FLOPs correlate with energy use, and (ii) better compatibility and easier deployment on edge devices. However, Henderson et al. [a] experimentally show that there is little correlation between FLOPs and energy usage when comparing across different model architectures. As PRO-SCALE has a different architecture than the default Mask2Former, there is no guarantee that the reduced FLOPs of PRO-SCALE will translate to reduced energy use. As reduced energy usage is also not shown experimentally, this benefit cannot be verified. As for compatibility and deployment on edge devices, the paper provides no evidence or references for these benefits, so these benefits can also not be verified.

Overall, while also considering the strengths of the work and the other reviews, these weaknesses cause me to keep my original rating.

[a] Henderson et al., "Towards the Systematic Reporting of the Energy and Carbon Footprints of Machine Learning," JMLR 2020.

---

> ### Author Response · Authors · 2024-11-19
> **Rebuttal response to Reviewer zhiy**
>
> Thank you Reviewer zhiy for your time to thoroughly evaluate our paper and your valuable feedback. It has greatly raised the quality of our manuscript. Please feel free to add any further questions based on our responses.
>
> ----
>
> - **Response to Weakness 1.** M2F's encoder system (or pixel decoder) consists of two parts- *a transformer encoder* that produces enhanced multi-scale tokens and a set of *convolutional layers* that produce the per-pixel embedding map. PRO-SCALE and the LPE module play crucial roles in optimizing computational efficiency of these components:
>     - In original M2F, multi-scale features from the backbone are processed through transformer layers in the encoder, which can be computationally expensive due to the global attention mechanism. PRO-SCALE significantly reduces this computational cost by introducing progressive token scaling, which adaptively reduces the number of tokens (and hence computations) at each stage while preserving key information.
>     - The pixel embedding map is generated using convolutional layers, which enhance local spatial details. However, the convolutional layers require processing an enormous number of tokens, leading to extremely high computational demands. With a simple structure of the LPE module, we show that by weakening the inductive bias due to the convolutional layers, the model still maintains high-quality results without taking on the computational burden.
>
>     This discussion has been added in the manuscript.
>
> ----
>
> - **Response to Weakness 2.** The stride is set to 1. The kernel size is set to 3 (provided in L249 of the submission version). These are the default parameters of the Maxpool2D operation which we didn’t change. We have added these details to the paper.
>
> ----
>
> - **Response to Weakness 3.** Our aim was to ensure the main contribution of the paper— progressively expanding
> tokens along the encoder depth to address the computation redundancy—remains clear and central to the reader. To avoid potential confusion, we did not include the token recalibration operation in the main figure but explicitly mentioned it in L233-236 and provided a detailed illustration in Figure 7 of the Appendix. We have added a note for the reader to refer to this figure for a complete visualization.
>
> ----
>
> - **Response to Weakness 4.** In the submission version, we mentioned in L790 that the TRC unit only costs 0.006 GFLOPs (computed on the COCO).
>
> ----
>
> - **Response to Weakness 5.** Thank you for the suggestion. We added FPS analysis for PRO-SCALE and other methods in Tab. 15 and Tab. 16.
>
> ----
>
> - **Response to Weakness 6.** We agree with the reviewer that with larger backbones, such as SWIN-L, backbone computations dominate. To address this, one can switch to a smaller backbone like SWIN-T to reduce backbone costs and incorporate PRO-SCALE to further optimize encoder computations, making the overall model more efficient.
>
> ----
>
> - **Response to Weakness 7.** We have updated the abstract to clarify that the GFLOPs reduction is for the encoder.
>
> ----
>
> - **Response to Weakness 8.** We have updated these tables to show the GFLOPs of the entire model.
>
> ----
>
> - **Response to Weakness 9.** ReMaX is a training algorithm designed to train small (or efficient models) and improve their ability to create better panoptic masks without any additional parameters. It doesn’t increase the efficiency but the performance. The GFLOPs of ReMaX trained models shown in Tab 1 and 2 have been updated now.
>
> ----
>
> - **Response to Weakness 10.** We agree with the reviewer that SL has lesser degradation than MoBy when compared to M2F. However, we wanted to say that integrating PRO-SCALE with the MoBY pre-trained backbone results in better performance compared to using SL backbone weights. We have modified the statement to reflect this.
>
> ----
>
> - **Response to Weakness 11.** Thank you so much for pointing these out. We have corrected all the typos in the revised paper.
>
> ----
>
> - **Response to Weakness 12.** Thank you. We have updated the caption of Tab. 3.
>
> ----
>
> - **Response to Weakness 13.** Thank you. We have updated all the tables to make the decimals consistent.
>
> ----
>
> - **Response to Question 1.** Small-resolution features encode compact, highly semantic information, whereas high-resolution features primarily capture fine-grained local details. In Appendix B, we illustrate that the abundance of high-resolution features often results in redundant information. Updating these features less frequently has minimal impact on performance while significantly reducing the computational cost associated with the attention mechanism. Hence, PRO-SCALE progressively adds the higher resolution features at later layers, thereby prioritizing semantic refinement in most layers. This reduces the number of query tokens, enabling a more computationally efficient multi-scale encoder.
>
> ----

---

> > ### Comment · Reviewer_zhiy · 2024-11-22
> > **Response to rebuttal**
> >
> > In the authors’ response, most of my concerns have been addressed. However, some of them still remain, and I have a few follow-up questions.
> >
> > ---
> >
> > **Regarding weakness 1:** I thank the authors for providing more detailed descriptions. However, the paper and the rebuttal still do not explain how M2F generates the pixel embedding map in detail. How many convolutional layers are used? What other operations are used? As a result, it is not fully clear what the exact differences are between the LPE module and the existing method.
> >
> > ---
> >
> > **Regarding weakness 5:** I thank the authors for providing some additional inference speed results in Tab. 15 and Tab. 16. However, Tab. 15 contains different model configurations than Tab. 2. In Tab. 2, PRO-SCALE(2,2,2) and PRO-SCALE(3,3,3) are shown to outperform Lite-M2F in terms of both PQ and GFLOPs. However, Tab. 15 only reports the FPS for PRO-SCALE(3,1,1) and PRO-SCALE(1,3,1), which obtain a considerably lower PQ. As a result, it is not clear how the FPS of PRO-SCALE(2,2,2) and PRO-SCALE(3,3,3) compares to Lite-M2F, and if these configurations are truly more efficient. Could the authors provide the FPS for PRO-SCALE(2,2,2) and PRO-SCALE(3,3,3)?
> >
> > Notably, with the results currently presented in Tab. 15, there is no benefit of using PRO-SCALE instead of Lite-M2F. In this table, Lite-M2F obtains a better PQ than PRO-SCALE while achieving a similar prediction speed. This limits the value of PRO-SCALE.
> >
> > Additionally, I still believe the paper would be considerably stronger if the FPS results were also provided for the ablations in Tab. 3. This would show the impact of the different design choices on the inference speed. Could the authors provide these results?
> >
> > ---
> >
> > **Regarding weakness 6:** I thank the authors for acknowledging that the effectiveness of PRO-SCALE diminishes for larger backbones. The authors' argument that this can be addressed by switching to a smaller backbone like Swin-T is not a strong argument to me. If one wants to achieve the best possible segmentation performance, then opting for a small backbone instead of a larger one is typically not an option, as this limits the performance.
> >
> > ---
> >
> > **Regarding weakness 10:** I thank the authors for the clarification. However, the new statement in the paper (L456-L458) is still incorrect, as it still refers to this “performance degradation”:
> >
> > > On average, integrating PRO-SCALE with the MoBY pre-trained backbone results in less performance degradation compared to using SL backbone weights, especially in instance segmentation (Fig. 5, right).
> >
> > As the authors acknowledged in the rebuttal: “SL has lesser degradation than MoBy when compared to M2F”. Therefore, this text need to be altered, e.g., by using a statement similar to the one provided by the authors in the rebuttal: “integrating PRO-SCALE with the MoBY pre-trained backbone results in better [overall] performance compared to using SL backbone weights.”
> >
> > ---
> >
> > **Regarding question 1:** I thank the authors for providing an answer to my question, but I am not convinced by the answer. The authors state that “the abundance of high-resolution features often results in redundant information” and that “updating these features less frequently has minimal impact on performance”. On their own, these statements are supported with experimental results. However, it is still not clear to me how these two things are related. **Why** does the presence of redundant information mean that these features can be updated less frequently with minimal impact on performance?
> >
> > ---
> >
> > I look forward to reading the authors’ response. Thanks in advance!

---

> ### Author Response · Authors · 2024-11-23
> **Response to further queries of Reviewer zhiy**
>
> Thank you Reviewer zhiy for your time and raising further questions. Please feel free to add any further questions based on our below responses.
>
> ----
> - **Response to comment 1 "difference between LPE module and the existing method".**
>     - The main focus of LPE module is not proposing a more efficient alternative to convolutions, but rather demonstrating that the original pixel embedding layer can be replaced with LPE—a lightweight, non-convolutional module—while maintaining competitive accuracy. This finding is significant because it advances the segmentation model architecture by identifying and addressing inefficiencies in the original design.
>     - At the architecture level, the differences in the embedding layer are the following:
>         - The original M2F ‘s embedding layer has the following structure: `lateral_conv (feature_in_channel, conv_dim, kernel_size=1) -> output_conv (conv_dim, conv_dim, kernel_size=3)`. In this, the largest computations come from the `output_conv (conv_dim, conv_dim, kernel_size=3)` layer.
>         - Our embedding layer has the following structure: `lateral_conv (feature_in_channel, conv_dim, kernel_size=1) -> LPE (conv_dim, conv_dim, kernel_size=3)`
>
>         Here, `lateral_conv` and `output_conv` are detectron2's Conv2D [A] operations. Following M2F, `conv_dim` is set to 256 and `feature_in_channel` is the number of channels of highest resolution feature map $s_1$. Both the convolutional operations and LPE modules are followed by normalization and non-linearity.
>
>     [A] https://detectron2.readthedocs.io/en/latest/modules/layers.html#detectron2.layers.Conv2d
>
> ----
> - **Response to further query on FPS.**
>     - **FPS for PRO-SCALE (2,2,2) and PRO-SCALE (3,3,3)**. As per reviewer's request, the FPS of PRO-SCALE (2,2,2) and PRO-SCALE (3,3,3) are provided in the table below (with SWIN-T backbone on Cityscapes).
>     | Method       | PQ | FPS   |
> | ------------ | -------- | ----- |
> | Lite-M2F     | 62.29    | 6.01 |
> | Ours (1,1,1) | 60.58    | 6.47 |
> | Ours (2,2,2) | 62.37    | 6.07 |
> | Ours (3,3,3) | 63.06    | 5.71  |
>     - **PRO-SCALE vs Lite-M2F benefits.** PRO-SCALE is better than Lite-M2F, as can be seen in the following results.
>         - As seen in the table above, PRO-SCALE clearly betters Lite-M2F both in performance and FPS (eg. configuration (2,2,2) provides better results with similar FPS).
>         - On a larger and diverse dataset like COCO, PRO-SCALE outperforms Lite-M2F significantly. For instance as shown in Tab. 1 with SWIN-T backbone, Lite-M2F provides a PQ of 52.70 with 31.81% encoder GFLOPs reduction whereas PRO-SCALE (3,3,3) results in a PQ of 52.82 with 51.99% encoder GFLOPs reduction.
>         - PRO-SCALE shows better performance than Lite-M2F performance both in segmentation and detection tasks as shown in Tab. 8 with lesser compute cost.
>     -  **FPS results in Tab. 3.** Based on the reviewer’s suggestion, we updated Tab. 3 with FPS values (last column) in the revised manuscript.
>
> ----

---

> > ### Comment · Reviewer_zhiy · 2024-11-25
> > **Response to authors**
> >
> > Thanks to the authors for providing further clarifications.
> >
> > After reading the authors’ responses, I still have two considerable remaining concerns.
> >
> > ---
> >
> > **Performance compared to Lite-M2F.** In the latest results provided by the authors (in Tab. 15 of the revised manuscript), PRO-SCALE(2,2,2) achieves a PQ of 62.37 at 6.07 FPS and Lite-M2F achieves a PQ of 62.29 at 6.01 FPS. In the author response, the authors state:
> >
> > >  PRO-SCALE clearly betters Lite-M2F both in performance and FPS (eg. configuration (2,2,2) provides better results with similar FPS).
> >
> > However, the differences in results between these methods are so small (0.08 PQ and 0.06 FPS) that I would not call them significant, e.g., the PQ difference might simply be due to randomness in the training process. As a result, there is no clear benefit of PRO-SCALE over existing method Lite-M2F.
> >
> > In their response, the authors also state that PRO-SCALE outperforms Lite-M2F significantly on COCO:
> >
> > >  For instance as shown in Tab. 1 with SWIN-T backbone, Lite-M2F provides a PQ of 52.70 with 31.81% encoder GFLOPs reduction whereas PRO-SCALE (3,3,3) results in a PQ of 52.82 with 51.99% encoder GFLOPs reduction.
> >
> > Again, however, (a) the PQ difference is not significant, and (b) it is not clear if PRO-SCALE(3,3,3) is actually faster than Lite-M2F. On the Cityscapes dataset, PRO-SCALE(3,3,3) is even slightly slower than Lite-M2F, see Tab. 15. To validate the claim that PRO-SCALE is really more efficient (and therefore better) than Lite-M2F on COCO, this should also be shown in terms of FPS.
> >
> > Finally, the authors state:
> >
> > > PRO-SCALE shows better performance than Lite-M2F performance both in segmentation and detection tasks as shown in Tab. 8 with lesser compute cost.
> >
> > However, again, the efficiency in terms of FPS is not shown, making it unclear if this “lesser compute cost” actually leads to speedups when the model is run, and if PRO-SCALE is truly more efficient than Lite-M2F (called Lite-DETR in Tab. 8).
> >
> > ---
> >
> > **Efficiency impact of LPE.** In the paper, the authors claim the LPE module as one of the main contributions. It replaces one convolutional layer with an average-pooling layer. In their response, the authors state the following:
> >
> > > The main focus of LPE module is not proposing a more efficient alternative to convolutions, but rather demonstrating that the original pixel embedding layer can be replaced with LPE—a lightweight, non-convolutional module—while maintaining competitive accuracy.
> >
> > However, the newly provided FPS results in Tab. 3 of the revised manuscript show that the impact of using the LPE module on the model’s prediction speed is very limited. While using the LPE module reduces the GFLOPs from 73.49 to 55.14 (25% decrease), the FPS only increases from 6.36 to 6.47 (2% increase). Moreover, using the module introduces a 1.15 PQ drop. In other words, while the efficiency impact in terms of GFLOPs may seem large, it is almost negligible in terms of actual prediction speed, while also leading to a performance drop. Therefore, this LPE module has little value in practice.
> >
> >
> > ---
> >
> > Because the main purpose of the paper is to improve efficiency while keeping the segmentation performance as high as possible, I believe the value of the paper is considerably limited when (a) the proposed method is not shown to obtain a better *prediction speed vs. segmentation performance* balance than a highly related existing method (Lite-M2F/Lite-DETR), and (b) the actual efficiency improvement (in terms of FPS) of one of the main contributions (LPE module) is almost negligible while causing a drop in segmentation performance.
> >
> > Overall, while also considering the strengths of the work and the other reviews, these weaknesses cause me to keep my original rating.

---

> ### Author Response · Authors · 2024-11-23
> **(Continued) Response to further queries of Reviewer zhiy**
>
> - **Point regarding larger backbones.** We appreciate the reviewer’s view-point here. We have the following comments.
>     - **Clarification on effectiveness of PRO-SCALE on large backbones.** PRO-SCALE is **effective** for M2F models with **both** large and small backbones. Regardless of the backbone size, it significantly reduces the encoder cost without causing a notable performance drop.
>         - With larger backbones like Swin-L, PRO-SCALE has a huge impact on the encoder's efficiency (50.87% reduction in table below). However, the total GFLOPs may not decrease dramatically (7.48% reduction) because the backbone's compute cost still dominates the overall model cost.
>         - In contrast, with smaller backbones like Swin-T, the encoder accounts for a larger proportion of the total compute budget. Hence, PRO-SCALE's impact on the encoder (51.96% reduction in table below) reduces the overall model compute cost significantly (27.09% reduction). \
>     To demonstrate the above two points quantitatively, we summarized the results from Tab. 2 and Tab. 7 below.
> | Backbone               | Original PQ | With PRO-SCALE PQ | Encoder Cost Reduction | Total Cost Reduction |
> |-------------------------|-------------|--------------------|-------------------------|-----------------------|
> | Smaller Backbone (SWIN-T) | 64.00      | 63.06             | 51.96%                 | 27.09%               |
> | Larger Backbone (SWIN-L)  | 66.60      | 65.93             | 50.87%                 | 7.48%                |
>     - **Response to *If one wants to achieve the best possible segmentation performance, then opting for a small backbone instead of a larger one is typically not an option, as this limits the performance.*** Please note that our core focus is to balance efficiency and performance in cost constrained settings, where smaller backbone excel (and with small backbones, encoder cost is the highest for the whole M2F model). Of course, larger backbones based M2F models excel in high segmentation quality, however, they fall short in resource-constraint scenarios. We summarize these two points in the table below.
> | Aspect                        | Small Backbones based M2F                                                                                    | Large Backbone based M2F                                            |
> |-------------------------------|-------------------------------------------------------------------------------------------------------|------------------------------------------------------------|
> | Performance-Accuracy Tradeoff | - Balanced accuracy and efficiency.                                                                  | - Achieves high segmentation accuracy, but poor tradeoff.                     |
> | Deployability                 | - Scales well across devices with low memory and compute needs.                                      | - Less suited for setups prioritizing compute cost.        |
>
>     **To conclude**, if we require to use M2F in a resource-constrained environment, opt for a smaller backbone and use PRO-SCALE for the encoder to achieve optimal performance.
>
> ----
> - **Comment on MoBY vs SL weights.** Thank you for the suggestion and we agree with the reviewer. We have modified the statement in the paper with “integrating PRO-SCALE with the MoBY pre-trained backbone results in better (overall) performance compared to using SL backbone weights.”
>
> ----
>
> - **Response to question *Why does the presence of redundant information mean that these features can be updated less frequently with minimal impact on performance?***. Here's some additional description.
>     - As shown in Fig. 9, low-resolution tokens ($s_4$, $s_3$​) demonstrate less redundancy, while high-resolution tokens ($s_2$​) shows more. The original M2F's encoder refines multi-scale features, concatenated along the token dimension to form $S=[s_4,s_3,s_2]$, by using deformable transformer attention [B]. Deformable transformer attention mechanism ensures that each token attends only to a small, learned set of keypoints across scales instead of the entire feature map. However, refining $S$ across all six encoder layers incurs significant computational cost.
>     - Now, since deformable attention requires only a few keypoints from each scale *and* $s_2$ is highly redundant, prioritizing frequent updates to less redundant features ($s_4, s_3$​ which contain compact information and rich semantics for most details of the scene [C]) maintains performance similar to the original model but drastically reduces computational cost.
>
>     Hence, in PRO-SCALE, $s_2$ is only introduced in layer set $p_3$ whereas $s_4$ is updated all layer sets $p_1$, $p_2$, and $p_3$.
>
>
>     [B] *Deformable DETR: Deformable Transformers for End-to-End Object Detection, ICLR 2021*  \
>     [C] *Lite DETR : An Interleaved Multi-Scale Encoder for Efficient DETR, CVPR 2023*
>
> ----

---

> > ### Author Response · Authors · 2024-11-23
> >
> > Please feel free to add any further questions based on our responses. We will be happy to alleviate further concerns.

---

> ### Author Response · Authors · 2024-11-26
> **Response to Reviewer zhiy**
>
> Hi Reviewer zhiy,
>
> Thank you for extensively engaging in discussions during this rebuttal period. We understand the reviewer wants to retain their score. We would like to highlight the following.
>
> **Note**: Our work targets reducing computational complexity (FLOPs), not FPS, a hardware-dependent metric. FPS reduction is orthogonal to our focus, and we will address this in our future works. The reviewer have themselves stated that **our work is valuable because it critically examines an inefficient component of a frequently used model, and finds that it can be made considerably more efficient while keeping the performance roughly the same**.
>
> ----
> -  **Performance w.r.t. Lite-M2F.**
>     - **Measuring using FLOPs**: The reviewer agrees with us that PRO-SCALE is better than Lite-M2F when considering GFLOPs (PRO-SCALE provides 51.99% reduction while Lite-M2F provides 31.81% reduction) while providing higher quality PQ (PRO-SCALE (3,3,3) PQ of 52.82 vs Lite-M2F PQ of 52.70). **Clearly, PRO-SCALE is more preferable.**
>     - **Measuring using FPS**: Based on the reviewer's request, we did an FPS analysis. With the point that **we did not aim to optimize for FPS efficiency in our work**,
>         - PRO-SCALE is again better than Lite-M2F even when considering FPS (PRO-SCALE provides 6.07 FPS while Lite-M2F provides 6.01 FPS) while providing higher quality PQ.
>         - The gains are agreeably minor, but **again, PRO-SCALE is still more preferable.**
>
>     **Conclusion:** Given the above two points, it is respectfully incorrect to say that *"As a result, there is no clear benefit of PRO-SCALE over existing method Lite-M2F"*, when the quantitative evidence suggests otherwise. This is further in contradiction to our range of ablation studies of application of PRO-SCALE involving different types of backbone  (Tab. 5), different types of weight initialization of the backbones (Fig. 5), different scales of backbones (Tab. 7), and different types of tasks that go beyond the universal segmentation (detection in Tab. 8, open-vocab segmentation in Tab. 10 and multi-task prediction in Tab. 11).
> ----
> -  **Contribution of LPE.**
>     - **What we claim:** The main focus of LPE module is not proposing a more efficient alternative to convolutions, but rather demonstrating that the original pixel embedding layer can be replaced with LPE—a lightweight, non-convolutional module—while maintaining competitive accuracy.
>     - **How we validated:** We showed using GFLOPs that it reduces the complexity of the model further by 25% decrease. The reviewer rightly points out that *the efficiency impact in terms of GFLOPs is large*. Even with FPS (again not our focus), this module provides a better speed.
>
>     **Conclusion:** Performance maintenance is evaluated from the perspective of the entire model, including PRO-SCALE. LPE focuses solely on reducing computational inefficiencies in design, and it effectively and objectively achieves this goal (25 % FLOP reduction and 2% better speed).
> ----
> We again appreciate and thank the reviewer dedicating their valuable time in discussing with us. We hope the reviewer supports our work. We will focus on improving FPS improvement in our future works.

---

> > ### Comment · Reviewer_zhiy · 2024-11-28
> > **Response to authors**
> >
> > Considering the authors’ latest response, I would like to make some clarifications.
> >
> > Like in my initial review, I acknowledge that PRO-SCALE achieves a reduction in FLOPs. However, from the initial submitted paper, it was not clear whether this FLOPs reduction also led to improved inference speed. In my view, decreasing FLOPs is not very useful if it doesn’t lead to improved model speed. If such a speedup is not achieved, what is the point of having a low number of FLOPs? For this reason, I specifically requested inference speed results.
> >
> > These inference speed results showed that (a) PRO-SCALE performs very similarly to existing method Lite-M2F in terms of both speed and segmentation performance (meaning that, to me, there is no clear benefit of PRO-SCALE over Lite-M2F), and (b) the proposed LPE module has very little impact on the speed while reducing the segmentation performance.
> >
> > Again, it is clear that PRO-SCALE requires fewer FLOPs than Lite-M2F, but what is the benefit of requiring fewer FLOPs if the prediction speed is roughly the same? Similarly, for the LPE module, a 25% FLOPs reduction does not have a significant benefit if it only leads to a 2% speed improvement.
> >
> > I am aware that FLOPs are hardware-independent and prediction speed is hardware-dependent, but I would like to note that some operations that require many FLOPs can simply be executed very efficiently and quickly by modern hardware. For example, the $3\times3$ convolution that the authors replace by avg-pooling in the LPE module can be executed very efficiently by GPUs, as shown by the fact that replacing the convolution with avg-pooling only improved the speed by 2%. As a result, I believe it is not appropriate to only look at FLOPs, but that we should additionally consider inference speed when assessing a model's efficiency.
> >
> > To conclude, I do not disagree with the authors that PRO-SCALE achieves significant FLOPs reductions, I just believe that these FLOPs reductions have limited value when they do not yield inference speed improvements, i.e., compared to existing work (Lite-M2F) or for one of the main contributions (LPE module). Therefore, I keep my rating unchanged.

---

> ### Author Response · Authors · 2024-11-29
> **Response to Reviewer zhiy's comment "If such a speedup is not achieved, what is the point of having a low number of FLOPs?"**
>
> Hi Reviewer zhiy,
>
> We appreciate your discussion on the significance of FLOPs reduction in connection to inference speed, and we would like to address the points you have raised, especially "*if such a speedup is not achieved, what is the point of having a low number of FLOPs?*". We start by acknowledging the validity of your request for inference speed comparisons. However,
>
>  - It **is not fully accurate to suggest that reducing FLOPs has no value** unless it translates directly to faster inference speed, especially on high-performance GPUs. While our experimental results indicate that FPS improvements on such GPUs may be limited, **lowering the number of FLOPs is fundamentally important** for several reasons **even if** immediate FPS gains are not achieved on  "modern hardware":
>     - **Energy Efficiency**: FLOPs correlate with energy use [D]. Reducing FLOPs enables more energy-efficient models, essential for large-scale and low-power deployments.
>         - The reviewer may have overlooked the additional axis of efficiency that GFLOPs reduction offers beyond FPS.
>     - **Deployment on Edge Devices and Compatibility with Hardware Accelerators**: These devices cannot use modern high-end GPUs, and hence require fewer FLOPs to ensure feasibility.  Fewer FLOPs improve compatibility with hardware like mobile devices and specialized processors, enhancing deployment on constrained systems.
>         - The reviewer appears to focus solely on high-end GPUs, but this overlooks the broader applicability of a model designed to run effectively on lower-end or specialized hardware, where FLOP reductions can play a crucial role.
>
>     *Conclusion*: PRO-SCALE's significantly lower FLOPs compared to baselines is a strong benefit beyond high-end GPU scenarios such as energy efficiency and feasibility with low-compute devices, and we show this with strong quantitative evidence via GFLOPs reduction.
>
> We appreciate the opportunity to engage in this discussion and thank you again for highlighting these points. Your feedback has allowed us to improve our paper (especially from the first feedback) and provide a more thorough understanding of our contributions.
>
> [D] *Trends in AI inference energy consumption: Beyond the performance-vs-parameter laws of deep learning* Sustainable Computing: Informatics and Systems, 38, 100857, 2023.

---

### Official Review · Reviewer_1znQ · 2024-10-31

**Soundness:** 3
**Presentation:** 3
**Contribution:** 2
**Rating:** 6
**Confidence:** 4

**Summary:**

This paper proposes the “PRO-SCALE” strategy to reduce the computational cost of Mask2Former encoders. The proposed approach can significantly reduce computations with minimal sacrifice in performance.

**Strengths:**

1. The writing and motivation of this paper is very clear.
2. The proposed method seems to achieve a better trade-off than several recent related approaches.

**Weaknesses:**

1. Although the authors emphasize the computational burden of the transformer encoder and propose dedicated methods to reduce computations, the overall reduction in proportion is not very obvious. The descriptions in the abstract seem to overstate the effect. It is better to clarify that 52% GFLOPs reduction in computation is for the encoder.

2. A figure to intuitively show the comparisons in trade-off (speed vs performance or FLOPs vs performance) is necessary for understanding the practical value of the approach.

3. The LPE module also contributes a lot to the reduction in computation. However, it is mainly because the computational efficiency of vanilla convolutions is too low. This further indicates that the proposed PRO-SCALE is not so important for the reduction of computational complexity of the entire system. Strictly speaking, the pixel embeddings generation module is not part of the transformer encoder. Compared to the LPE module, the TRC module in the appendix seems more interesting.

**Questions:**

Why not use depth-wise 3x3 convolutions for the LPE module?

---

> ### Author Response · Authors · 2024-11-19
> **Rebuttal response to Reviewer 1znQ**
>
> Thank you Reviewer 1znQ for your time to evaluate our paper and your valuable support. Please feel free to add any further questions based on our responses.
>
> ----
>
>
> - **Comment on the abstract.** We have updated the abstract to clarify that the GFLOPs reduction is for the encoder.
>
> ----
>
> - **Figure to show the FLOPs vs performance trade-off.** We have added Fig.10 in the appendix to show the GFLOPs vs performance  trade-off.
>
> ----
>
> - **Comment on impact of PRO-SCALE and LPE module.**
>     - We note that M2F calls the whole encoder system as “pixel decoder”. The pixel decoder provides two outputs to the next stage (i.e. the decoder): refined multi-scale features (using the transformer encoder) and the pixel embedding map (using the convolutional layers). Hence, we follow their system to design PRO-SCALE and the LPE module within this encoder system.
>         - **an example**: In Tab. 3, it can be observed that if we replace the original M2F's encoder system with PRO-SCALE's (1, 1, 1) configuration (no LPE), the cost goes from 281 GFLOPs to 132 GFLOPs (reduction of 149 GFLOPS). If we further replace the pixel embedding layer with LPE module, the total cost goes from 132 GFLOPs to 73.49 GFLOPs (reduction of 58.51 GFLOPS). Clearly, it indicates that PRO-SCALE is extremely important for the reduction of computational complexity of the entire system as it results in more GFLOPs reduction.
>         - Also note that, the main focus of LPE is not proposing a more efficient alternative to convolutions, but rather demonstrating that the original pixel embedding layer can be replaced with LPE—a lightweight, non-convolutional module—while maintaining competitive accuracy. This finding is significant because it advances the segmentation model architecture by identifying and addressing inefficiencies in the original design.
>     - To avoid potential confusion on the main contribution of our paper, we did not include the token recalibration operation in the main manuscript due to the minor performance difference.
>
> ----
>
> - **Question on using depth-wise 3x3 convolutions in LPE.** Our main focus is to show that the pixel encoder in the M2F architecture can be greatly lightened, rather than improving the efficiency of the convolution itself. While the reviewer’s suggestion of using depth-wise 3x3 convolutions aligns with the need to reduce computations, it would still introduce a larger computational burden compared to our proposed LPE module.
> ----

---

> > ### Comment · Reviewer_1znQ · 2024-11-25
> > **Response to rebuttal**
> >
> > Thank authors for your response. But I still believe there is no substantial difference between using a depth-wise convolutional layer or a pooling layer in this context. I will keep my scores.

---

### Official Review · Reviewer_g4Mz · 2024-11-02

**Soundness:** 3
**Presentation:** 3
**Contribution:** 3
**Rating:** 6
**Confidence:** 4

**Summary:**

The paper introduces an efficient transformer encoder architecture that progressively increases token length and feature scale, coupled with a LPE module. This approach achieves an effective balance between computational efficiency and segmentation quality.

**Strengths:**

1. The paper presents its ideas with clarity and good technical writing.
2. The experimental validation is comprehensive, featuring:
    - Thorough comparative analyses
    - Well-structured ablation studies
    - Clear demonstration of each component's contribution to overall performance

**Weaknesses:**

1. In Table9 for FPS comparison, while Lite-M2F and RT-M2F are used as baselines in other evaluations, a complete comparison with all baseline models on FPS would strengthen the efficiency claims
2. The paper states that ReMaX is orthogonal to the approach and ineffective on larger models, but could you provide quantitative evidence? Currently, the table results only demonstrate ReMaX has a good performance.

**Questions:**

For Figure6, could you provide more visualization comparisons with also some baselines like Lite-M2F, RT-M2F, or PEM?

---

> ### Author Response · Authors · 2024-11-19
> **Rebuttal response to Reviewer g4Mz**
>
> Thank you Reviewer g4Mz for dedicating time to evaluate our paper and your valuable support. Should there be any additional questions or feedback regarding our responses, please do not hesitate to raise further queries.
>
> ----
>
> - **FPS comparison with Lite-M2F and RT-M2F.** Based on the reviewer's suggestions, we analyzed the FPS of Lite-M2F and RT-M2F against PRO-SCALE in the table below. PRO-SCALE achieves an optimal/better balance by delivering a high PQ score alongside strong FPS performance compared to other methods, ensuring an effective trade-off between quality and speed. PRO-SCALE has the distinct advantage to adjust its $(p_1, p_2, p_3)$ configuration to adapt to user's FPS vs performance requirements.
> | Method       | Performance (PQ) | FPS   |
> | ------------ | ----------------------------------- | ----- |
> | Lite-M2F     | 62.29                               | 6.01 |
> | RT-M2F       | 59.73                               | 6.59 |
> | **Ours** (1,1,1) | 60.58    | 6.47 |
> | **Ours** (2,2,2) | 62.37    | 6.07 |
> | **Ours** (3,3,3) | 63.06    | 5.71  |
>
> ----
>
> - **Comparison to ReMaX.** ReMaX is a training algorithm designed to improve the panoptic mask quality of small or efficient models without introducing additional learnable parameters. While it enhances performance, **it does not improve efficiency**. As noted in Appendix C of the ReMaX paper, "the improvement of ReMaX gets saturated when the (model parameter) numbers become high." For instance, Table 11 in the ReMaX paper shows only a 0.3-0.6% PQ improvement on the kMaX-DeepLab baseline when applied to ConvNeXt-T and ConvNeXt-S backbones. In contrast, our method focuses on reducing encoder computations across all model sizes, providing a more consistent trade-off between efficiency and performance.
>
>
> ----
>
> - **Additional visualization comparisons.** Based on the reviewer's suggestion, we added additional visualizations in the Appendix for Lite-M2F and PEM in Fig. 11 of the revised manuscript.
>
> ----

---

> > ### Author Response · Authors · 2024-11-26
> > **Any further clarifications?**
> >
> > Hi Reviewer g4Mz,
> >
> > Thank you again for reviewing our work and insightful comments. We wanted to check if our responses have clarified your questions. Your support is extremely valuable to us.
> >
> > Best wishes, \
> > Authors

---

> > > ### Comment · Reviewer_g4Mz · 2024-11-28
> > >
> > > Thank you for the detailed reply. Your responses addressed most of my concerns. Compared to Lite-M2F, the improvement is not very significant, but it offers flexible choices. I will keep my current score.

---

> > > > ### Author Response · Authors · 2024-11-29
> > > > **Thank you Reviewer g4Mz!**
> > > >
> > > > Thank you Reviewer g4Mz for confirming that our responses have addressed most of your concerns. We appreciate your support.
> > > >
> > > > Best Wishes, \
> > > > Authors

---

### Official Review · Reviewer_GmJD · 2024-11-10

**Soundness:** 3
**Presentation:** 3
**Contribution:** 3
**Rating:** 6
**Confidence:** 4

**Summary:**

This paper proposes to Progressive Token Length Scaling, a strategy designed to enhance the efficiency of transformer encoders in universal segmentation models. The authors identify that the state-of-the-art model, Mask2Former, devotes over 50% of its computational resources to the transformer encoder, largely because it processes a full-length token representation from all backbone feature scales at every encoder layer. This paper addresses this inefficiency by progressively scaling the token length with each encoder layer, thereby significantly reducing computational demands—specifically, achieving around 52% reduction in GFLOPs—while maintaining performance levels on the COCO dataset.

**Strengths:**

The progressive integration of finer-grain information within the transformer encoder is an intuitive and straightforward approach that proves to be both simple and effective.

A comprehensive set of experiments on segmentation and detection tasks validate the design choices of the PRO-SCALE architecture, demonstrating its effectiveness.

The paper is clearly written and easy to follow, enhancing its accessibility and understanding.

**Weaknesses:**

None

**Questions:**

What distinguishes the LPE module from traditional fully convolutional network (FCN) style upsampling methods?

If I understand correctly, the computational costs in P1 and P2 are quadratically lower compared to P3. Would it be advantageous to use a different number of layers for each split instead of maintaining the same number of layers across all splits?

---

> ### Author Response · Authors · 2024-11-19
> **Rebuttal response to Reviewer GmJD**
>
> Thank you Reviewer GmJD for taking the time to review our paper and for your valuable support. Please feel free to raise further questions based on our responses, if any.
>
> ----
>
> - **Difference w.r.t. traditional fully convolutional network (FCN) style upsampling methods.** The LPE module differs from traditional FCN-style upsampling methods primarily in its computational efficiency and design philosophy. Traditional FCNs use fully convolutional layers for pixel embedding, which require processing large input sizes, leading to significant GFLOPs and high computational costs. In contrast, the LPE module avoids these costly learnable layers, relying instead on lightweight operations to achieve similar functionality. By leveraging simpler mechanisms for pixel embedding, the LPE module enhances local detail and provides high-quality masks without adding computational overhead to the encoder. This design ensures a better balance between accuracy and efficiency, particularly for tasks requiring pixel-level detail.
>
> ----
>
> - **Using different number of layers for each split.** We assume that the reviewer may be suggesting the use of a different number of layers for each split. The reviewer is correct on the conclusion that the computational costs in $p_1$ and $p_2$ are lower compared to $p_3$  (but since M2F uses deformable attention for the encoder layers, it is linear). As shown in Tab. 6 of the manuscript, we can definitely use a different number of layers for each split. Setting the  ($p_1$, $p_2$, $p_3$) configuration values are independent of each other. The advantage is: different ($p_1$, $p_2$, $p_3$) configurations can allow multiple options for the encoder for the same computational budget.
>
> ----

---

> > ### Author Response · Authors · 2024-11-26
> > **Any further clarifications?**
> >
> > Hi Reviewer GmJD,
> >
> > Thank you again for reviewing our work and positive comments. We wanted to check if our responses have clarified your questions. Your support is extremely valuable to us.
> >
> > Best wishes, \
> > Authors

---

> > > ### Comment · Reviewer_GmJD · 2024-12-03
> > >
> > > Thank you to the authors for addressing my questions. While the measured FPS improvement is only marginal, which has prompted me to slightly adjust my rating downward, I still consider this a valuable and well-executed contribution overall.

---

> > > > ### Author Response · Authors · 2024-12-03
> > > >
> > > > Hi Reviewer GmJD,
> > > >
> > > > Thank you for your thoughtful feedback and for acknowledging our efforts in addressing your questions. We deeply appreciate your recognition of the value and execution of our contribution.
> > > >
> > > > As we have discussed with other reviewers, the FPS metric was not a primary optimization target for our work. Our focus was on reducing the model's computational cost (GFLOPs), which offers significant advantages such as improved energy efficiency and suitability for low-compute devices.
> > > >
> > > > Best Wishes, \
> > > > Authors

---

### Meta-Review · Area_Chair_4GoR · 2024-12-19

**Metareview:**

The paper proposes PRO-SCALE, an efficient transformer encoder for universal segmentation that significantly reduces computational cost while maintaining performance, and introduces the Light Pixel Embedding (LPE) module to optimize the pixel embedding process within the Mask2Former architecture.

During rebuttal and discussions, three reviewers keep the original ratings and one reviewer slightly adjusts the score downward to 6, leading to the final scores of 6,6,6,5. The remaining biggest concern is limited inference speed improvement despite of significant FLOPs reduction.

After carefully  reviewing the paper and all the reviewers' comments and discussions, The AC agrees that the actual inference speed improvement is crucial for practical usage. However, Considering that this paper presents a solid method  with extensive evaluations and clear explanations, the AC recognizes its significant value in the compression for segmentation models. With an average score of 5.75, the paper is competitive for this year's ICLR conference. The AC agrees to accept the paper and strongly recommends incorporating the content from the rebuttal into the final version.

**Additional Comments On Reviewer Discussion:**

- Reviewer GmJD acknowledges the simplicity and effectiveness of PRO-SCALE and appreciates the clear presentation of the paper, while also noting the importance of energy efficiency and deployment on edge devices due to the reduced GFLOPs, even if the FPS improvement is marginal.
- Reviewer g4Mz commends the paper's clarity and thorough experimental validation, including well-structured ablation studies, but suggests that a complete comparison with all baseline models on FPS would strengthen the efficiency claims and requests quantitative evidence for the claim regarding ReMaX's ineffectiveness on larger models.
- Reviewer 1znQ points out the value of the proposed method in achieving a better trade-off than several recent related approaches and questions the significance of the GFLOPs reduction, suggesting that the LPE module's contribution to computational efficiency might be overstated and that a depth-wise convolutional layer could be a more efficient alternative.
- Reviewer zhiy provides a comprehensive evaluation, recognizing the method's simplicity and effectiveness, while also raising concerns about the actual efficiency improvements, the impact of the LPE module on prediction speed, and the need for more substantial evidence to support certain claims, such as the relationship between redundancy in feature tokens and the PRO-SCALE design.
- The authors defend their method's efficiency and practicality, emphasizing the importance of reducing GFLOPs for energy efficiency and compatibility with low-compute devices, and clarify that the LPE module's purpose is to demonstrate the inefficiencies in the original pixel embedding layer design rather than to propose a more efficient alternative to convolutions.

---

### Decision · Program_Chairs · 2025-01-22

Accept (Poster)